



# Constraining the geothermal parameters of *in situ* Rb–Sr dating on Proterozoic shales and its subsequent applications

Darwinaji Subarkah[1,6], Angus L. Nixon[2,6], Monica Jimenez[3], Alan S. Collins[1,6], Morgan L. Blades[1], Juraj Farkas[4,6], Sarah E. Gilbert[5], and Simon Holford[3]

[1]Tectonics & Earth Systems (TES), Department of Earth Sciences, University of Adelaide, Adelaide, SA 5005, Australia

[2]Apatite Thermochronology Lab and Services (ATLAS), Department of Earth Sciences, University of Adelaide, Adelaide, SA 5005, Australia

[3]Stress, Structure and Seismic, Australian School of Petroleum and Energy Resources (ASPER), University of Adelaide, Adelaide, SA 5005, Australia

[4]Metal Isotope Group (MIG), Department of Earth Sciences, University of Adelaide, Adelaide, SA 5005, Australia

[5]Adelaide Microscopy, University of Adelaide, Adelaide, SA 5005, Australia

[6]MinEx CRC, Australian Resources Research Centre, Perth, WA 6151, Australia

*Correspondence to*: Darwinaji Subarkah (Darwinaji.subarkah@adelaide.edu.au)

**Abstract** Recent developments in tandem laser ablation-mass spectrometer technology have been shown to be capable of separating parent and daughter isotopes of the same mass online. As a result, beta decay chronometers can now be applied to the geological archive *in situ* as opposed to through traditional whole-rock digestions. One novel application of this technique is the *in situ* Rb–Sr dating on Proterozoic shales that are dominated by authigenic clays. This method can provide a depositional window for shales by differentiating signatures of early diagenetic processes versus late-stage secondary alteration. However, the hydrothermal sensitivity of the Rb–Sr isotopic system across geological timescales in shale-hosted clay minerals is not well understood. As such, we dated the Mesoproterozoic Velkerri Formation from the Altree 2 well in the Beetaloo Sub-basin (greater McArthur Basin) using *in situ* Rb–Sr geochronology and constrained its thermal history using common hydrocarbon maturity indicators, and modelled effects of contact heating due to the intrusion of the Derim Derim Dolerite.

*In situ* Rb–Sr dating of mature, oil-prone shales in the diagenetic zone from the Velkerri Formation in this study yielded ages of 1470 ± 102 Ma, 1457 ± 29 Ma, and 1421 ± 152 Ma. These results agree with previous Re–Os dating of the unit and are interpreted as recording the timing of an early diagenetic event soon after deposition. Conversely, overmature, gas-prone shales in the anchizone sourced from stratigraphically deeper within the same borehole and succession were dated at 1318 ± 105 Ma and 1332 ± 67 Ma. These ages are younger than the expected depositional window for the Velkerri Formation. Instead, they are consistent with the age of the Derim Derim Dolerite mafic intrusion intersected 800 m below the Velkerri Formation. Thermal modelling suggests that a single intrusion of 75 m thickness would have been capable of producing a significant hydrothermal perturbation radiating from the sill top. The intrusion width proposed by this model is consistent with similar Derim Derim Dolerite sill thicknesses





found elsewhere in the McArthur Basin. The extent of the hydrothermal aureole induced by this intrusion coincide

with the point in which kerogen from the Velkerri Formation becomes overmature. As a result, the mafic intrusion

intersected here is interpreted to have caused kerogen in these shales to enter the gas window, induced fluids that

mobilise trace elements and resetting the Rb–Sr chronometer. Consequently, we propose that the Rb–Sr

chronometer in shales may be sensitive to temperatures of ca. 110°C in hydrothermal reactions but can withstand

temperatures of more than 190°C in thermal systems absent of fluid. Importantly, this study demonstrates a

framework for the combined use of *in situ* Rb–Sr dating and kerogen maturation indicators to help reveal the

thermochronological history of Proterozoic sedimentary basins. As such, this approach can be a powerful tool for

identifying the hydrocarbon potential of source rocks in similar geological settings.

## 1 Introduction

The Rb–Sr isotopic system has historically been one of the most powerful dating tools in Earth science. Rb is

abundant in K-rich minerals such as micas, clays, and K-feldspar and these minerals are commonly found in a wide

range of geological settings (Simmons, 1998). As such, it is an effective technique to date processes including

igneous emplacement, metamorphism, sedimentation, and hydrothermal alteration (Nebel, 2014). Its long half-life

also makes it applicable to date events as early as the infant stages of our solar system (Minster et al., 1979; Nebel et

al., 2011; Papanastassiou and Wasserburg, 1970). Traditionally, the application of this method required an arduous

process of column chromatography to chemically separate the parent ($^{87}$Rb) and daughter ($^{87}$Sr) isotopes and avoid

isobaric interference between the two elements (Charlier et al., 2006; Dickin, 2018; Faure, 1977; Hahn et al., 1943;

Hahn and Walling, 1938; Yang et al., 2010). Alas, this approach has historically been expensive, time consuming,

and results in the loss of the genetic relationships between the minerals analysed which caused the technique to lose

its popularity in recent years (Nebel, 2014).

Recent advancements in tandem laser ablation inductively coupled plasma mass spectrometry (LA-ICP-MS/MS)

and similar instruments have revitalised the use of Rb–Sr by allowing them to be applied *in situ* (Bevan et al., 2021;

Gorojovsky and Alard, 2020; Hogmalm et al., 2017; Redaa et al., 2021a; Yim et al., 2021; Zack and Hogmalm,

2016). A reactive gas can be introduced into a reaction cell between quadrupoles in an LA-ICP-MS/MS system,

which permits the online separation of $^{87}$Sr from $^{87}$Rb through the measurement of the mass shifted Sr reaction

product (Hogmalm et al., 2017; Redaa et al., 2021a; Zack and Hogmalm, 2016). This allows for a more rapid and





economic analysis time as well as the ability to preserve petrographic relationships during these analyses. Consequently, secondary input of Rb or Sr from inclusions, zonation, alteration, and detritus can be isolated, resulting in a better understanding of the geochronological results. The application of similar setups with other beta-decay dating systems have also yielded promising results (Harrison et al., 2010; Hogmalm et al., 2019; Ribeiro et al., 2021; Simpson et al., 2021; Tamblyn et al., 2021).

As a result, the *in situ* Rb–Sr dating method can now be used very similarly to laser ablation U–Pb dating, where age information can be obtained reliably, rapidly, and cheaply. Datable minerals for this method are often macroscopically visible and abundant, avoiding the need for extensive mineral separation commonly needed for U–Pb geochronology (Zack and Hogmalm, 2016). In addition, the initial $^{87}Sr/^{86}Sr$ ratio from the calculated isochron and the elemental data concurrently collected with the Rb and Sr isotopes can fingerprint the geochemical nature of the samples analysed (Li et al., 2020; Redaa et al., 2021b; Subarkah et al., 2021; Tamblyn et al., 2020). This approach have been shown to be capable of dating paragenetic sequences in deformation structures (Armistead et al., 2020; Tillberg et al., 2020), hydrothermal alteration assemblages (Laureijs et al., 2021), magmatic and metamorphic events (Li et al., 2020; Tamblyn et al., 2020) as well as metallogenic systems (Olierook et al., 2020; Redaa et al., 2021b) whilst still preserving their micro-scale textural context.

Another novel use of this technique is to date Proterozoic shales in order to constrain their depositional window (Subarkah et al., 2021). Evidence suggests that clay minerals in Proterozoic shales are dominated by authigenic products from reverse weathering processes during reactions in equilibrium with the water-column (Isson and Planavsky, 2018; Kennedy et al., 2006; Mackenzie and Kump, 1995; Rafiei and Kennedy, 2019; Rafiei et al., 2020). Conversely, clay assemblages in late Ediacaran and Phanerozoic shales are commonly dominated by detrital products from continental weathering of soils (Baldermann et al., 2020; Chamley, 1989; Galán, 2006; Hillier, 1995; Kennedy et al., 2006; Rafiei et al., 2020; Singer, 1980; Wilson, 1999). Simple multicellular organisms such as fungi and lichen have been shown to dramatically influence the rate of chemical weathering in continental rocks (Chen et al., 2000; Cuadros, 2017; Kennedy et al., 2006; Lee and Parsons, 1999; McMahon and Davies, 2018; Mergelov et al., 2018; Rafiei et al., 2020). As such, the surge in abundance of detrital clays shales in the Ediacaran and Phanerozoic has been attributed to the production of soils driven by emergence of these microorganisms (Kennedy et al., 2006; Lee and Parsons, 1999; McMahon and Davies, 2018; Mergelov et al., 2018; Rafiei and Kennedy, 2019;



Zambell et al., 2012). Thus, the primarily authigenic nature of clay minerals in Proterozoic shales make them ideal targets for *in situ* Rb–Sr dating (Subarkah et al., 2021).

Despite the promising potential of the Rb–Sr isotopic system, the chronometer still holds some limitations. Rb and Sr are large ion lithophile elements that can sit in well-bound interstitial sites within a mineral lattice as well as adsorbed onto the surface where they are more susceptible to fluid mobilisation (Li et al., 2019; Nebel, 2014; Villa, 1998). In these environments, fluid-induced recrystallisation and alteration can drive element and isotopic exchange at lower effective closure temperatures than those empirically determined for classic thermal volume diffusion

reactions (Dodson, 1973; Field and Råheim, 1979; Jenkin et al., 1995; Villa, 1998). Nevertheless, these weaknesses can in turn be used advantageously to date secondary events such as episodes of hydrothermal fluid-flow (Dodson, 1973; Li et al., 2020; Redaa et al., 2021b; Shepherd and Darbyshire, 1981; Subarkah et al., 2021; Tamblyn et al., 2020).

In this study, we show how *in situ* Rb–Sr analysis of the Proterozoic Velkerri Formation in the Roper Group,

McArthur Basin, northern Australia (Figure 1) dated clay-mineral recrystallisation at temperatures similar to those at which kerogen catagenesis occurs. The Roper Group is a good case study for *in situ* Rb–Sr shale dating as it has been shown to be dominated by authigenic clays (Rafiei and Kennedy, 2019; Subarkah et al., 2021) and is chronologically well-constrained (Ahmad and Munson, 2013; Bodorkos et al., 2020; Cox et al., 2022; Kendall et al., 2009; Southgate et al., 2000; Subarkah et al., 2021; Yang et al., 2020). Furthermore, the resurgence of interest in the

resource potential of the organic-rich Velkerri Formation has also yielded a framework of palaeotemperature data that aim to discern the maturation history of hydrocarbons within the unit (Ahmad and Munson, 2013; Capogreco, 2017; Cox et al., 2016; Crick et al., 1988; George and Ahmed, 2002; Jarrett et al., 2019; Lemiux, 2011; Revie, 2014; Summons et al., 1994; Taylor et al., 1994; Volk et al., 2005).

Here, we targeted the Velkerri Formation (Figure 1) from the thoroughly investigated well Altree 2 (Capogreco,

2017; Cox et al., 2022; Cox et al., 2016; George and Ahmed, 2002; Jarrett et al., 2019; Lemiux, 2011; Nguyen et al., 2019; Nixon et al., 2021; NTGS, 1989, 2009, 2010, 2012; Revie, 2014; Sander et al., 2018; Yang et al., 2018). We show that common hydrocarbon maturation proxies such as $T_{max}$ data from Rock-Eval pyrolysis and illite crystallinity can help define the temperature sensitivity of the Rb–Sr isotopic system in organic-rich shales. In addition, we have also modelled the geothermal aureole of a mafic intrusion that may have matured the kerogen into





the gas window, altered trace elemental signatures and reset the Rb–Sr isotopic system within the unit. As a result, we demonstrate that combining this novel dating method with traditional kerogen maturation proxies is a powerful tool for reconstructing the thermochronological evolution of Proterozoic basin systems and can aid in hydrocarbon exploration in similar settings.

## 2 Geological Background

The Palaeo-to-Mesoproterozoic greater McArthur Basin is an intra-cratonic sedimentary system exposed across 180,000 km$^2$ of northern Australia (Ahmad and Munson, 2013). The basin is sub-divided into five unconformity-bounded sedimentary packages characterized by similarities in age, lithology, and stratigraphic position (Jackson et al., 1999; Rawlings, 1999). The Roper Group is part of the Wilton Package which is the youngest of these sub-divisions (Jackson et al., 1999; Jackson et al., 1987; Rawlings, 1999). The thickness of the Roper Group varies

around 1 to 5 km across several different fault zones (Abbott and Sweet, 2000; Abbott et al., 2001; Ahmad and Munson, 2013; Jackson et al., 1987; Rawlings, 1999). The Beetaloo Sub-basin (Figure 1) is interpreted to be the main depocentre of the sedimentary system and the Roper Group is thickest here (Abbott and Sweet, 2000; Ahmad and Munson, 2013; Jackson et al., 1987; Plumb and Wellman, 1987). Lithologically, the Roper Group comprises of a series of coarsening-up sequences dominated by marine mudstone and interbedded sandstone with minor

successions of intraclastic limestone (Abbott and Sweet, 2000; Ahmad and Munson, 2013; Jackson et al., 1987; Yang et al., 2018). Records of water-column euxinia and redox stratification as well as fluctuating salinity levels suggests that the Roper Group formed in an intermittently restricted marine basin within an epicontinental setting similar to the modern Black Sea, or Baltic Sea (Ahmad and Munson, 2013; Beyer et al., 2016; Cox et al., 2022; Cox et al., 2016; Mukherjee and Large, 2016; Revie and MacDonald, 2017; Yang et al., 2018).

Age constraints of the Roper Group have been established through several geochronological methods (Ahmad and Munson, 2013; Jackson et al., 1999; Kendall et al., 2009; Nixon et al., 2021; Page et al., 2000; Southgate et al., 2000; Subarkah et al., 2021; Yang et al., 2019; Yang et al., 2020; Yang et al., 2018). The beginning of the group's genesis is bracketed by a SHRIMP U–Pb zircon study from a tuff within the unconformably underlying Nathan Group as well as minimum depositional age from an *in situ* Rb–Sr analysis in the lower Roper Group that yielded

ages of 1589 ± 3 Ma and 1577 ± 56 Ma, respectively (Page et al., 2000; Subarkah et al., 2021). The unconformity between the Roper Group and the older Nathan Group is likely related to the Isan Orogeny ca. 1.58 Ga (Ahmad and





Munson, 2013; Jackson et al., 1999). Absolute dating of the Roper Group has been obtained through two SHRIMP

U–Pb zircon studies from tuff layers in the Mainoru Formation resulting in ages of 1492 ± 4 Ma and 1493 ± 4 Ma

(Jackson et al., 1999; Southgate et al., 2000). On the other hand, the Kyalla Formation at the top of the Roper Group

is constrained to having being deposited between the U–Pb age of its youngest detrital zircon at 1313 ± 47 Ma

(Yang et al., 2018) and the age of crosscutting Derim Derim dolerite intrusions at 1313 ± 1 Ma (Yang et al., 2020).

The Velkerri Formation within the Roper Group is the focus of this study and mature organic-rich shales from this

unit have been dated by Re–Os analysis at 1417 ± 29 Ma and 1361 ± 21 Ma (Kendall et al., 2009). These have been

interpreted to be the depositional age of the formation. The geochronological constraints of the Roper Group are

summarized in Figure 1. The Velkerri Formation is dominated by deep-basinal lithologies such as black mudstones

and silts that coarsens-up into the cross-bedded Moroak Sandstone and Sherwin Ironstone (Abbott et al., 2001). The

Velkerri Formation is interpreted to represent a deep-water, high-stand systems tract within a marine environment

(Abbott et al., 2001; Warren et al., 1998). The Velkerri Formation is commonly sub-divided into three distinct

Upper, Middle and Lower Members based on their variations in total organic carbon (TOC) content, gamma ray

response, geochemistry, sedimentology and mineralogy (Ahmad and Munson, 2013; Cox et al., 2016; Cox et al.,

2019; Munson and Revie, 2018; Revie, 2016; Warren et al., 1998).

Importantly, the McArthur Basin experienced a complex thermal history following the deposition of the Wilton

Package. Mafic sills of the Derim Derim Dolerite widely intrude all units in the Roper Group at ca. 1330–1295 Ma,

with the oldest intrusions likely contemporaneous with the end of sedimentation in the basin (Ahmad and Munson,

2013; Bodorkos et al., 2020; Nixon et al., 2021; Subarkah et al., 2021; Yang et al., 2020). Little evidence of

subsequent tectono-thermal perturbation is present within the basin until much of the region was overlain by

subaerial basaltic lavas of the Kalkarindji Large Igneous Province (LIP) extruded at ca. 510 Ma (Evins et al., 2009;

Glass and Phillips, 2006; Jourdan et al., 2014). Following the Cambrian expulsion of the Kalkarindji lavas, no

significant (> 110°C; Nixon et al., submitted) heating  has been detected within the shallow parts of the basin

(Duddy et al., 2004).

The Altree 2 well drilled in the Beetaloo Sub-basin is chosen for this study as it intersects the entirety of the Velkerri

Formation (Figure 2). In addition, the well terminated an intrusion of the Derim Derim Dolerite and also intersected

lavas of the Kalkarindji LIP that directly overlie the Proterozoic sedimentary rocks. Importantly, this well has also



been the focus of numerous geochronological, geochemical and geobiological investigations from academia, private

explorers as well as the Northern Territory Geological Survey (NTGS) which provide important complementary

data to supplement this study (Bodorkos et al., 2020; Cox et al., 2022; Cox et al., 2016; Cox et al., 2019; George and

Ahmed, 2002; Jarrett et al., 2019; Lemiux, 2011; Nguyen et al., 2019; Nixon et al., 2021; NTGS, 2009, 2010, 2012;

Sander et al., 2018; Warren et al., 1998; Yang et al., 2018).

## 3 Methodology

Rock-Eval pyrolysis, bulk x-ray diffraction (XRD) mineralogical and well log data were collated from several

sources and compiled together in this study (Capogreco, 2017; Cox et al., 2016; Lemiux, 2011; NTGS, 1989, 2009,

2010, 2012; Revie, 2014). As such, their corresponding methodologies can be found in the references therein. The

lithology of the Velkerri Formation was interpreted in detail (Figure 2) using the electrical logs Gamma-Ray (GR),

Neutron (NRS) and Density (RHOB) of the Altree-2 well (NTGS, 1989). Four lithologies were defined after

applying cut-offs at each electrical log. They are then correlated along depth. Sandstone units corresponds to a GR <

API, NRS < 0.20 % and RHOB of around 2.5 gr/cm$^3$. This relates to a cross over between the RHOB and NPRS

logs and competent material at the GR. Interbedded shale and sands are defined by a GR > 130 and < 250 API, NRS

> 0.20 and < 0.25 m$^3$/m$^3$ and RHOB between 2.5 and 2.53 gr/cm$^3$. This lithology reflected a smaller breach between

the density and neutron logs in comparison to the previous sandstone lithology. Shale units were constrained by a

185 GR > 250 API, Neutron > 0.25 m$^3$/m$^3$ and RHOB > 2.53 gr/cm$^3$ and a minimum to no separation between the

porosity logs. On the other hand, dolomitic siltstones have a GR response similar to the sandstone, with NRS

ranging between 0.25 to 0.27 and RHOB > 2.62 gr/cm$^3$. This indicates a competent lithology in the GR with a gap

between the neutron and density curves. In addition, $T_{max}$ data was also collated to discriminate the hydrocarbon

maturation levels downhole. From this, a shift in hydrocarbon potential and $T_{max}$ gradients were identified at around

190 900 m (Figure 2), were kerogen enters the gas window and becomes overmature. Five shale chips were then

sampled from the Velkerri Formation in Altree 2 at depths 415 m, 520 m, 696 m, 938 m, and 1220 m for further

characterisation.

Samples were first imaged for their mineral composition and petrographic relationships. Back Scatter Electron

(BSE) imaging and Mineral Liberation Analysis (MLA) maps of samples were collected using a Hitachi SU3800

Automated Mineralogy Scanning Electron Microscope at Adelaide Microscopy. BSE image tiles were done at 10





mm working distance and 20kV acceleration voltage with MLA maps were completed using a raster analysis with spectra collected at 0.35 µm/pixel resolution. Minerals previously categorised by bulk XRD analysis of the Velkerri Formation from Cox et al. (2016) were used to develop a 'library' to help identify phases found by spectral reflectance MLA mapping. *In situ* Rb–Sr geochronology and trace element analysis were undertaken at Adelaide Microscopy using a laser ablation (RESOlution-LR ArF 193nm excimer laser) inductively coupled plasma tandem mass spectrometer (Agilent 8900x ICP-MS/MS) with the analytical parameters and tuning conditions following Redaa et al. (2021a) , which also provides details of the specific laser setup. Laser ablation data were processed using the LADR software package (Norris and Danyushevsky, 2018) and isochron ages were calculated with ISOPLOTR (Vermeesch, 2018). The $^{87}$Rb decay constant used was 0.000013972 ± 4.5e-7 Myr$^{-1}$ following Villa et al. (2015).

The phologopite nano-powder Mica-Mg (Govindaraju et al., 1994) was used as the primary reference material and its natural mineral equivalent MDC from Bekily Mine in Madagascar (Armistead et al., 2020; Hogmalm et al., 2017; Li et al., 2020; Redaa et al., 2021a) as well as glauconite reference material GL-O (Charbit et al., 1998; Derkowski et al., 2009) were used secondary age standards. As previously discussed in Subarkah et al. (2021), nano-powdered reference materials have similar ablation characteristics to fine-grained shales, with analogous matrix effects. As such, they are ideal standards for *in situ* analyses of these samples. When anchored to a $^{87}$Sr/$^{86}$Sr initial ratio = 0.72607 ± 0.00363 as reported by Hogmalm et al. (2017), MDC yielded an age of 524 ± 7 Ma. This is within error of the published mean age of Mica-Mg at 519 ± 7 Ma (Hogmalm et al., 2017). In addition, the independent reference material GL-O gave an age of 96 ± 4 Ma, accurate to its published age of 93 ± 4 Ma (Charbit et al., 1998; Derkowski et al., 2009). Glass standards NIST SRM 610 was used as a primary standard for elemental quantification in this study. Analysis of secondary standard BCR-2G yielded major, trace and rare earth element composition that were in good agreement (Pearson R > 0.999, Pearson R$^2$ > 0.999, and P Value < 0.0001) with their published values as compiled in the GeoREM database (Jochum and Stoll, 2008; Jochum et al., 2011; Jochum et al., 2005; Pearce et al., 1997).

One-dimensional thermal modelling of the Altree 2 well was conducted using the SILLi 1.0 numerical model, which is designed for simulating thermal perturbation associated with sill emplacement within sedimentary basins (Iyer et al., 2018). First, palaeotemperatures were estimated from the compiled T$_{max}$ data (Disnar, 1986, 1994; Waliczek et




al., 2021) following equations based on similar sedimentary systems that experienced a heating event due to burial
and a subsequent igneous intrusion (Piedad-Sánchez et al., 2004). Forward modelling was then conducted to

replicate maximum thermal conditions calculated in the well from $T_{max}$ data, where palaeotemperatures suggest the
Upper and Middle Velkerri Formation experienced significant additional sedimentary cover present during the
Mesoproterozoic. During modelling, an additional 1 km of sedimentary rocks were added above the erosional
unconformity now present above McArthur Basin, while all post-Mesoproterozoic units were excluded. The upper
contact of a sill with an initial temperature of 1150°C (Wang et al., 2012) ) was set at 2368 m, in accordance with

adjusted burial depths during the Mesoproterozoic. As sill thickness is unconstrained within the Altree 2 well,
multiple iterations were run with different thicknesses in order to establish the scenario able to best satisfy the
thermal aureole extent observed in this well. From this, a sill thickness of 75 m was considered most appropriate,
and is consistent with Derim Derim sill thicknesses of ~10–100 m commonly observed across the basin (Lanigan
and Ledlie, 1990; Lanigan and Torkington, 1991; Ledlie and Maim, 1989; NTGS, 2014, 2015, 2016). Full modelling

parameters and petrophysical properties are provided in the Supplementary Material Table S2 and Table S3..

## 4 Results

### 4.1 Compilation of legacy data

All legacy data are compiled in the Supplementary Material and were checked for quality before interpretation as
several factors such as contamination of cuttings due to drilling fluid or poor organic content can make results

unreliable (Carvajal-Ortiz and Gentzis, 2015; Dembicki Jr, 2009; Peters, 1986). As such, Rock-Eval pyrolysis values
were screened using the thresholds described by Hall et al. (2016). Data were subsequently excluded from
interpretation if these criteria were not met. More than 90% of the data yielded S2 > 0.1 mg HC/g, indicating that
they were sufficiently abundant in organic content to provide well-defined peaks for characterising $T_{max}$ and
Hydrogen Index. Importantly, compilation of Rock-Eval pyrolysis values were all internally consistent (e.g.,

Hydrogen Index = S2/TOC x 100). Next, there was no evidence of anomalously low $T_{max}$ values (< 380°C) present.
Extremely low $T_{max}$ values are commonly a product of incorrect selection of the S2 peak by the program or the
widening of the S1 peak from non-indigenous free hydrocarbons. $T_{max}$ results compiled in this study ranges between
384°C to 502°C with a mean of 433°C (st. dev. = 17). TOC content in the Velkerri Formation varies from 0.07% to
8.07%, averaging at 2.25% (st. dev. = 2.26).



Clay mineral crystallinity and size data sourced for this compilation were standardised for interlaboratory comparisons (Warr and Mählmann, 2015; Warr and Rice, 1994). Full-width at half maximum values were computationally remeasured as a secondary check (Capogreco, 2017; NTGS, 2010, 2012). Thirteen samples from the Velkerri Formation were analysed for their illite crystallinity. The Kübler Index for these shales range between 0.88 to 0.36, with decreasing values at depth and the lowest data originating from the Lower Velkerri Formation.

### 4.2 Mineralogy of the Velkerri Formation

Eleven mineral phases were identified by bulk XRD analysis of the Velkerri Formation from Cox et al. (2016). The major mineral phases were quartz, kaolinite, illite, and orthoclase, which on average make up 90% of the total composition of the samples. Trace minerals include glauconite, montmorillonite, pyrite, magnetite, siderite, dolomite, and plagioclase. Our MLA mapping also identified these assemblages. Importantly, the two different
methods categorised these minerals at similar abundances. However, results from MLA mapping also found other mineral assemblages not identified by bulk XRD analysis, including biotite, chlorite, clinochlore, apatite, and zircon. These differences could be due to the slightly different sub-intervals from which samples were analysed. Bulk XRD is a destructive procedure and therefore, the same section cannot be reused for *in situ* analysis. As a result, samples spaced 1 – 2 cm apart may yield different results. The complete mineralogical abundance and correlations between
results bulk XRD and MLA mapping are summarised in Table 1. Extensive petrographic descriptions of all samples can be found in the Supplementary Material.

### 4.3 Laser ablation data

Geochronological results yielded by samples from the Upper and Middle Velkerri Formation gave ages within error of each other. The sample from 415 m depth was dated at 1470 ± 102 Ma. Next, the mudstone analysed from 520 m
depth yielded an age of 1457 ± 29 Ma. Thirdly, the shale sample studied from 696 m depth resulted in an age of 1421 ± 152 Ma. A Lower Velkeri Formation shale chip from 938 m at depth was dated at 1318 ± 105 Ma. Another sample from this member, towards the boundary with the underlying Bessie Creek Sandstone at depth 1220 m resulted in an age of 1332 ± 67 Ma. The range of precision from these Rb–Sr ages are primarily constrained by a good spread in $^{87}Rb/^{86}Sr$ ratios, the number of data points defining the regression line as well as errors on each
individual analysis (Nebel, 2014). The most precisely dated samples, extracted from 520 m and 1220 m depth, had the widest range of $^{87}Rb/^{86}Sr$ ratios (0 – 50) whilst the other two samples preserved a range of $^{87}Rb/^{86}Sr$ values less





than 10 (Figure 6). The variability in these values could be a subject of future studies in order to improve the success of this dating method.

Elemental concentrations of each sample were concurrently collected during the *in situ* Rb–Sr laser ablation
investigation and they are in good agreement with data collected by bulk geochemical analysis from Cox et al. (2016). Samples from depth 415 m, 520 m, and 696 m do not show any covariation between their total REEY concentrations and Sm/Nd ratios (Figure 7). On the other hand, sample 938 m showed a statistically significant relationship between these two parameters (Pearson R = 0.58, Pearson $R^2$ = 0.336, P Value < 0.0001). In addition, Velkerri Formation shale sourced from depth 1220 m also showed a strong covariation between total REEY values
and Sm/Nd (Pearson R = -0.545, Pearson $R^2$ = 0.297, P Value < 0.0001). These associations were also identified in the whole-rock geochemical data collected from Cox et al. (2016). Figure 7B shows that samples between 390 – 900 m depth hold no statistically significant relationships between the two factors. However, samples from deeper than 900 m display a strong relationship between the two variables (Pearson R = -0.559, Pearson $R^2$ = 0.312, P Value = 0.003). The full geochronological and inorganic geochemical dataset for samples in this study can be found in the
Supplementary Material.

**4.4 Thermal Modelling**

One-dimensional thermal modelling of the emplacement of a 75 m thick Derim Derim Dolerite sill at the base of the Altree 2 well is sufficient to produce a thermal aureole reaching temperatures > 100°C, ca. 800 m above the top contact of the sill (Figure 8A). Maximum palaeotemperatures recorded in the upper Velkerri Formation exceed those
predicted in this simulation, however, this may be attributed to elevated temperatures in the shallow basin during eruption of the Kalkarindji LIP in the Cambrian (Nixon et al., submitted). The resultant maximum thermal profile is consistent with $T_{max}$ derived palaeotemperature estimates and is thus considered a plausible model for the observed data from the well. Post-intrusion temperatures at depths that match the samples with reset Rb–Sr ages are much lower than observed in comparable isotopic systems for thermally induced diffusion (Dodson, 1973; Tillberg et al.,
2020; Torgersen et al., 2015; Yoder and Eugster, 1955), with the shallowest reset sample peaking at ca. 110°C. Furthermore, elevated temperatures predicted by the modelling are geologically short lived, with temperatures returning to steady-state conditions by approximately half a million years following sill intrusion (Figure 8B).





| Depth | Method | Apatite | Biotite | Chlorite | Clinochlore | Dolomite | Glauconite | Illite | Kaolinite | Magnetite | Montmorillonite | Orthoclase | Pyrite | Quartz | Siderite | Zircon | Plagioclase |
|---|---|---|---|---|---|---|---|---|---|---|---|---|---|---|---|---|---|
| 415 m | Bulk XRD | 0.00 | 0.00 | 0.00 | 0.00 | 0.25 | 0.82 | 34.05 | 21.25 | 0.36 | 1.79 | 13.33 | 0.28 | 26.13 | 1.76 | 0.00 | 0.00 |
|  | MLA Map | 0.00 | 1.22 | 1.32 | 0.01 | 0.00 | 0.00 | 35.52 | 7.45 | 0.89 | 9.05 | 11.48 | 0.40 | 23.93 | 0.00 | 0.02 | 4.39 |
| 520 m | Bulk XRD | 0.00 | 0.00 | 0.00 | 0.00 | 0.37 | 2.35 | 43.28 | 17.14 | 0.38 | 2.01 | 10.01 | 0.74 | 23.67 | 0.05 | 0.00 | 0.00 |
|  | MLA Map | 0.00 | 0.07 | 0.00 | 0.08 | 0.00 | 0.00 | 36.51 | 7.11 | 0.00 | 0.04 | 4.08 | 0.24 | 47.94 | 0.00 | 0.00 | 0.85 |
| 696 m | Bulk XRD | 0.00 | 0.00 | 0.00 | 0.00 | 0.21 | 1.53 | 27.59 | 2.07 | 0.00 | 1.49 | 13.40 | 4.11 | 45.02 | 0.00 | 0.00 | 4.57 |
|  | MLA Map | 0.31 | 0.93 | 0.00 | 0.06 | 0.00 | 0.00 | 18.48 | 0.00 | 0.00 | 4.00 | 8.27 | 4.47 | 60.88 | 0.01 | 0.00 | 1.41 |
| 938 m | Bulk XRD | 0.00 | 0.00 | 0.00 | 0.00 | 0.53 | 1.36 | 27.03 | 6.99 | 0.33 | 1.16 | 11.32 | 1.49 | 43.28 | 0.00 | 0.00 | 6.50 |
|  | MLA Map | 1.2 | 0.00 | 0.00 | 0.06 | 0.00 | 0.00 | 22.13 | 5.46 | 0.00 | 0.00 | 3.37 | 1.60 | 57.79 | 0.00 | 0.00 | 6.90 |
| 1220 m | Bulk XRD | 0.00 | 0.00 | 0.00 | 0.00 | 0.43 | 1.04 | 39.36 | 13.99 | 0.42 | 1.66 | 7.32 | 0.26 | 33.56 | 0.10 | 0.00 | 1.86 |
|  | MLA Map | 0.00 | 0.01 | 0.00 | 0.01 | 0.05 | 0.00 | 38.18 | 20.21 | 0.00 | 0.00 | 4.45 | 0.01 | 35.25 | 0.00 | 0.01 | 1.63 |



| Depth (m) | Pearson R | Pearson R 95% C.I. | Pearson R$^2$ | P Value |
|---|---|---|---|---|
| 415 | *0.922* | *0.784–0.973* | *0.850* | *< 0.0001* |
| 520 | *0.866* | *0.648–0.953* | *0.750* | *<0.0001* |
| 696 | *0.953* | *0.868–0.984* | *0.910* | *<0.0001* |
| 938 | *0.965* | *0.898–0.988* | *0.930* | *<0.0001* |
| 1220 | *0.989* | *0.969–0.996* | *0.979* | *<0.0001* |

Table 1A. Mineralogical abundance of the Velkerri Formation shales collected by bulk XRD analysis from Cox et al. (2016) and spectral reflectance MLA mapping in this study. All values are in weight percent. B. Covariation between the mineral phases categorised by both methods.

## 5 Thermal Maturity of the Velkerri Formation

Geochemical and mineralogical-based thermal maturity indicators collected via Rock-Eval studies and bulk XRD analyses were compiled in this study in order to establish a vertical profile of the Velkerri Formation to assess the local palaeo-thermal structure. The $T_{max}$ parameter is the temperature at which the maximum rate of hydrocarbon generation occurs during pyrolysis analysis and is a common method used to reconstruct thermal histories of basin systems (Espitalié, 1986; Espitalié et al., 1977; Peters and Cassa, 1994; Welte and Tissot, 1984). Additionally, the Kübler index (KI) is determined by a sample's first X-ray diffraction reflection peak and is also a popular maturation proxy used to classify low-grade metamorphism in pelitic rocks (Blenkinsop, 1988; Guggenheim et al., 2002; Kubler, 1967). However, both of these thermal indicators can be influenced by multiple factors other than burial-related heating and therefore struggle to resolve absolute quantitative palaeotemperatures. Abundance in hydrogen, sulphur, and uranium content or the organic richness of samples can result in inaccurate $T_{max}$ values (Dembicki Jr, 2009; Espitalié et al., 1977; Peters, 1986; Yang and Horsfield, 2020). Similarly, the KI has also been shown to be sensitive to several parameters such as changes heating rate and geochemical variability in the sample's initial mineralogy (Abad and Nieto, 2007; Eberl and Velde, 1989; Mählmann et al., 2012; Warr and Mählmann, 2015). In addition, variations in procedures between laboratories can further complicate the direct comparison of these values (Cornford et al., 1998; Jarvie, 1991; Peters and Cassa, 1994; Tissot et al., 1987). Consequently, these thermal indicators need to be treated with caution when applied independently and are more suitable as qualitative



discriminators as opposed to absolute quantitative parameters. However, such proxies can be more confidently used to estimate palaeotemperatures in sedimentary successions if they show a strong relationship with each other (Burtner and Warner, 1986; Dellisanti et al., 2010; Ola et al., 2018; Velde and Espitalié, 1989; Waliczek et al., 2021). Ultimately, both organic and inorganic indicators are essential for a robust understanding of the thermal histories of sedimentary sequences through time.

In this study, we examine the covariation between the $T_{max}$ values and KI to reconstruct the thermal history of the

Velkerri Formation in the Altree 2 well (Figure 3). In our compilation, samples with immature kerogen ($T_{max} <$ 435°C) correspond to rocks in the diagenetic zone (KI > 0.45°Δ2θ). This relationship is true in similar studies and generally translates to palaeotemperatures of ca. 100°C (Abad and Nieto, 2007; Dellisanti et al., 2010; Espitalié et al., 1977; Kosakowski et al., 1999; Kubler, 1967).

Interestingly, the samples within the mature oil window (435°C < $T_{max}$ < 465°C) show a wide range of KI values

between 0.39 and 0.65°Δ2θ (Figure 3). This is possibly due to the delay between thermal reactions in clay minerals as opposed to organic matter (Ola et al., 2018). Although the maturation of organic matter and the morphology of clay minerals both largely depend on temperature, other processes such as the kinetics of the thermal reaction and geochemical composition of the sample can make these relationships non-linear (Meunier et al., 2004; Ola et al., 2018; Pollastro, 1993; Varajao and Meunier, 1995; Velde and Vasseur, 1992). The disparity between kerogen

evolution and the equilibrium stage of illitisation at these temperatures may also play a role in this variability (Dellisanti et al., 2010). Nevertheless, an increase in $T_{max}$ pyrolysis results from these samples still appears to correlate with a decrease in KI values. These thermometers would approximately equate to palaeotemperatures between 100 and 150°C (Árkai et al., 2002; Frey and Merriman, 1999; Kosakowski et al., 1999; Welte and Tissot, 1984).

Lastly, the sample displaying over-mature kerogen ($T_{max} >$ 465°C) understandably corresponds to the smallest KI value (Figure 3) of 0.36°Δ2θ (Dellisanti et al., 2010). These values commonly define the gas window and the anchizone, corresponding to palaeotemperatures of ca. 200°C (Árkai et al., 2002; Dellisanti et al., 2010; Kosakowski et al., 1999). Overall, a trend between increasing $T_{max}$ and decreasing KI values (Figure 3) confirms the feasibility of these parameters as thermal maturation proxies (Dellisanti et al., 2010).





These thermal parameters can be further examined by inspecting the changes in kerogen type collected by Rock-Eval pyrolysis analysis (Dellisanti et al., 2010; Espitalié et al., 1985; Espitalié et al., 1977). This can be defined by relationships between the hydrogen index (HI), oxygen index (OI), hydrocarbon potential, and TOC content (Banerjee et al., 1998; Cooles et al., 1986; Espitalié et al., 1985; Espitalié et al., 1977; Peters, 1986). As previously mentioned, $T_{max}$ values down-hole suggests that the organic matter within the Upper and Middle Velkerri formation

(depth 390 – 900 m) was within the oil window, reflecting an immature-to-mature source rock (Figure 2). However, there is a notable shift into the gas window occurring at ca. 900 m depth in the Lower Velkerri Formation (Cox et al., 2016; Lemiux, 2011; NTGS, 1989, 2009, 2010, 2012; Revie, 2014). This shift is mirrored by other Rock-Eval data used to determine kerogen types in the Velkerri Formation (Figure 4). Figure 4 shows that the Upper and Middle Velkerri Formation are dominated by immature-to-mature source rocks with oil prone Type II-III kerogen,

suggesting a palaeotemperature window of less than 150°C (Banerjee et al., 1998; Espitalié et al., 1985; Hunt, 1995; Welte and Tissot, 1984). Meanwhile, the Lower Velkerri Formation comprises of source rocks in the over-mature zone with gas-prone Type III kerogen. A subset of samples from the Lower Velkerri Formation plot in the dry gas window, suggesting that they may have reached palaeotemperatures of over 200°C (Dellisanti et al., 2010; Frey and Merriman, 1999; Hunt, 1995; Welte and Tissot, 1984).

Multiple geochemical and mineralogical thermal parameters from our compiled data set demonstrate good correlation between them, suggesting that the proxies used in this study primarily recorded changes in palaeotemperature as opposed to other possible variables. Notably, three different, independent, source-rock maturation proxies statistically agree with each other and recorded similar temperature intervals. As such, we investigated five samples approaching the geothermal anomaly in the Lower Velkerri Formation for *in situ* Rb–Sr

and trace element analysis. The changes in thermal maturation indexes throughout the well are used to help constrain the parameters of the Rb–Sr isotopic system in Proterozoic shales.

## 6 Thermochronological History of the Velkerri Formation

The abundant clays in samples from depth 415 m, 520 m, and 696 m are predominantly illite, with trace amounts of chlorite, kaolinite, and montmorillonite (Table 1). Notably, they do not show typical irregular, angular detrital

morphologies (Figure 5, Figure S2A-C). Instead, clay minerals in these samples form a matrix cement, filling in porous spaces, wrapping around detrital grains and suggesting that they formed within the sediment (Rafiei and



Kennedy, 2019; Rafiei et al., 2020; Subarkah et al., 2021). A minor component of illite can also be seen replacing micas and feldspars (Figure 5, Figure S2A-C). Primary sedimentary structures such compaction of clays along the bedding plane can also still be identified in these samples (Figure 5, Figure S2A-C and Supplementary Material).

These petrographic relationships are further discussed in the Supplementary Material and were similarly found in Roper Group shales elsewhere, indicating an early-diagenetic origin (Rafiei and Kennedy, 2019; Subarkah et al., 2021). Importantly, the ages from these samples are all in agreement with the deposition of the Velkerri Formation dated at 1417 ± 29 Ma by Re–Os geochronology (Kendall et al., 2009), suggesting that the analysed bulk-composition illite matrix of these samples formed not long after deposition.

Nevertheless, we also sought to identify any potential secondary alteration of these shales by analysing their geochemical signatures. Sm/Nd ratios are common geochemical proxies for screening alteration in shales as Nd is preferentially lost relative to Sm during post-depositional processes (Awwiller and Mack, 1989; Awwiller and Mack, 1991). In addition, fluid-rock reactions also have a significant impact on rare earth element and yttrium solubility and transportation during hydrothermal events (Lev et al., 1999; Williams-Jones et al., 2012). Therefore,

these parameters can be an effective tool for highlighting fingerprints of post-depositional geochemical mobilisation (Figure 7). Samples from depths shallower than 900 m show no significant relationships between their total rare earth elements and yttrium (REEY) concentrations and Sm/Nd ratios collected by laser ablation analysis or through traditional bulk trace element geochemistry (Cox et al., 2016). As such, the ages produced by these samples can be interpreted as a minimum depositional age for the unit and as having recorded an early diagenetic event as opposed

to a late-stage secondary overprint. Furthermore, $T_{max}$ values for the Velkerri Formation at depths 390 – 900 m were ca. 435°C, suggesting they are within the oil window (Figure 2). In addition, other pyrolysis parameters indicate that samples from this subset are dominated by oil-prone Type II kerogen (Figure 4). As a result, we propose that the geothermal activity that would have driven the hydrocarbons into such parameters are not sufficient to disturb the Rb–Sr and trace element systems in these shales.

Conversely, shales collected from depths > 900 m showed petrographic evidence of post-depositional alteration (Figure 5 and Figure S2D-E). Clay minerals in sample from 938 m depth are fissile and foliated. In addition, pyrite and apatite can be observed overgrowing illite and chlorite. Moreover, Lower Velkerri Formation shale from 1220 m notably display more crystalline morphologies (Figure 5 and Figure S2D-E). Illite aggregates in this sample



preserve features inconsistent with an early diagenetic origin (and Figure S2D-E). They interlock with quartz

overgrowth and appear to replace earlier clay matrices. Samples from these depths gave younger ages than those

from the subset stratigraphically above them (Figure 6). However, not only are these ages younger than the results

from the subset of samples above, they are also younger than the proposed deposition intervals acquired through the

Re–Os dating of the unit (Kendall et al., 2009). Instead, these ages are consistent with the emplacement of the mafic

Derim Derim Dolerite intersected at the base of the Altree 2 well at depth 1696 m. This intrusion has been dated

twice using U–Pb geochronology of baddeleyite and apatite, at 1313 ± 1 Ma (Nixon et al., 2021) and 1328 ± 24 Ma

(Yang et al., 2020), respectively. Consequently, the shale samples from depth below 900 m may have recorded a

secondary heating event induced by the emplacement of the Derim Derim Dolerite sill.

In addition to petrographic and geochronological disparities, samples from depths below 900 m display statistically

significant relationships between total REEY concentrations and Sm/Nd ratios (Figure 7). The shale chip analysed

from 938 m had a positive relationship between an increase in Sm/Nd ratio and total REEY concentration (Pearson

r: 0.580, R-squared: 0.336), while the sample collected from depth 1220 m preserved a negative relationship

between Sm/Nd ratio and total REEY values (Pearson r: -0.545, R-squared: 0.297). These associations are similarly

reflected in the bulk trace element data collated from Cox et al. (2016). In the compiled data, shales from deeper

than 900 m demonstrate a strong affinity between these controls (Pearson r: -0.559, R-squared: 0.312), whereas

samples above 900 m show no variation (Figure 7B). These alteration indicators are further evidence that the Lower

Velkerri Formation at depths below 900 m experienced a late-stage secondary heating event. This event may have a

hydrothermal component as trace and rare earth elemental systems are more readily mobilised in fluid systems

(Awwiller and Mack, 1989; Awwiller and Mack, 1991; Condie, 1991; Lev et al., 1999; Poitrasson et al., 1995;

Williams-Jones et al., 2012). Importantly, Rock-Eval pyrolysis data from this interval suggest that kerogen in these

shales are thermally overmature (Figure 2) and are dominated by gas-prone Type III source rocks. Previous studies

have shown that the source rocks in the Velkerri Formation were overmature only when affected by similar

magmatic events (Crick et al., 1988; George and Ahmed, 2002). As such, the Derim Derim Dolerite intrusion may

have prompted a hydrothermal alteration footprint onto its surrounding sediments via conductive heat loss and/or

heat-transfer fluids. This magmatic pulse would have recrystallised the minerals, disturbed the Rb–Sr chronometer,

mobilised trace elements, and heated the kerogen within the Lower Velkerri Formation to overmaturity.





Overall, the Upper and Middle Velkerri Formation preserve an early diagenetic event immediately after deposition of the unit. Petrographic relationships deduced through high resolution spectral reflectance imaging showed that clay minerals in these shales formed cements and grew in pore spaces, replacing micas and feldspars (Figure 5 and Supplementary Material). Furthermore, different geochemical indicators collected by laser ablation and whole-rock

analysis both suggest that they have not experienced extensive secondary hydrothermal alteration (Figure 7). Moreover, the geochronological results from these successions (Figure 6) agree with the proposed depositional age for the unit (Kendall et al., 2009). Temperature constraints for this section are within the oil window and diagenetic zone (Figure 2, 3, and 4).

Shale samples sourced from deeper than 900 m in Altree 2, however, preserve evidence for a late-stage,

hydrothermal event. Firstly, samples from this depth display foliated and crystalline morphologies that may indicate secondary processes. This is further evinced by the significant relationships shown by alteration proxies for samples from the Lower Velkerri Formation (Figure 7). The Rb–Sr ages yielded from this subset of samples are younger than the depositional ages for the unit dated through Re–Os geochronology (Kendall et al., 2009). Instead, they reflect the age of the Derim Derim Dolerite sourced at 1696 m down-hole (Nixon et al., 2021; Yang et al., 2020). As a result,

the secondary event experienced by the Lower Velkerri Formation is likely induced by the emplacement of this mafic intrusion. Further, this depth threshold is where multiple thermal maturation proxies (Figure 2, 3 and 4) suggest that these rocks have experienced temperatures in the anchizone and overmature gas window. These indicators suggest that source-rocks within this interval may have experienced palaeotemperatures of at least 150°C (Dellisanti et al., 2010; Frey and Merriman, 1999; Hunt, 1995; Welte and Tissot, 1984). Interestingly, this is in good

agreement with evidence from aqueous fluid inclusions in quartz veins within the Derim Derim Dolerite elsewhere which suggest that hydrocarbons from the Velkerri Formation migrated in the cooling sill at similar temperatures ca. 130°C (Dutkiewicz et al., 2004).

**7 Modelled Predictions of the Geothermal Aureole Induced by the Derim Derim Dolerite**

Resetting of Rb–Sr geochronology and overmaturation of hydrocarbons in the Lower Velkerri Formation within the

Altree 2 well implies the presence of a secondary thermal and/or hydrothermal aureole extending ca. 800 m away from the Derim Derim Dolerite sill, which as intersected at present day depth 1696 m. This event was responsible for the maturation of organic matter into the gas window, mobilisation of trace elements, and resetting the Rb–Sr





chronometer in the Lower Velkerri Formation. One-dimensional thermal modelling for a sill thickness of 75 m in

the Mesoproterozoic suggests temperatures exceeding the oil window over 120°C (Tissot et al., 1974; Waples, 1980)

only extended ca. 600 m from the intrusion (Figure 8A).

Samples at present day depths of 938 m and 1220 m yield Rb–Sr ages corresponding to emplacement timing of the

Derim Derim Dolerite (Nixon et al., 2021; Yang et al., 2020), which suggests that the intrusion caused the

chronometer to reset. Predicted temperatures experienced by the shallowest reset sample, however, are lower than

the inferred closure temperatures for observed K–Ar and Rb–Sr in sheet silicates (Dodson, 1973; Tillberg et al.,

2020; Torgersen et al., 2015; Yoder and Eugster, 1955). In a scenario in which a sill of thickness 75 m was intruded

below samples, rocks from present day depth of 938 m are only predicted to have experienced maximum heating to

ca. 110°C (Figure 8C), with temperatures exceeding 100°C for a duration of only ca. 50 ka (Figure 8B).

Additionally, the eruption of lavas from the Kalkarindji LIP (Evins et al., 2009; Glass and Phillips, 2006; Jourdan et

al., 2014), within the same vertical profile offer an intriguing opportunity to evaluate thermal resistance of the Rb–Sr

system in shale-hosted clays in different conditions. Basaltic lavas of the ca. 510 Ma Cambrian Kalkarindji LIP

(Evins et al., 2009; Glass and Phillips, 2006; Jourdan et al., 2014) are preserved above Proterozoic sedimentary

rocks in the Altree 2 well. Furthermore, regional apatite fission track data suggest that the thermal pulse induced

during this LIP extrusion were short-lived but sufficient (> 190°C) to anneal tracks in the upper ~500 m of the basin

(Nixon et al., submitted). However, the shallowest samples taken in this study (at depths 415 m, 520 m, and 696 m)

did not have their Rb–Sr isotopic system disturbed despite experiencing such temperatures from this reheating event.

Consequently, the thermal profile for the sample at 415 m depth provides a minimum closure temperature constraint

for short-lived conditions which have not reset the Rb–Sr chronometer in these (presumably) dry shales over 800

million years after the Derim Derim dolerite intrusion. Interestingly, the Cambrian palaeotemperatures imposed by

the Kalkarindji lavas (> 190°C; Nixon et al., submitted) are notably higher than Mesoproterozoic

palaeotemperatures reached by samples with Rb–Sr ages reset by the Derim Derim Dolerite (ca. 110°C; Figure 8A

and 8B).

Such disparity suggests that the presence of fluid (either connate, or sourced from the intrusion), rather than just

temperature, is likely to play a critical role in determining whether the Rb–Sr record in a shale is reset. As such, the

geochemical system in shales within the aureole may be disturbed at lower temperatures, as trace and rare earth



elements are more easily mobilised in hydrothermal fluid systems (Li et al., 2019; Nebel, 2014; Poitrasson et al.,

1995; Villa, 1998; Williams-Jones et al., 2012).

**8 Significance of Constraining *In Situ* Rb–Sr Dating on Proterozoic Shales**

Velkerri Formation shales intersected by the Altree 2 well preserved the presence of an elevated thermal gradient in

a ~800 m thick section overlying an intrusion of the Derim Derim Dolerite (Figure 8 and 9). *In situ* Rb–Sr isotopic

ages from the Upper and Middle Velkerri Formation above this hydrothermal aureole yielded ages (Figure 6)

overlapping with its depositional window (Kendall et al., 2009), unaltered trace element compositions (Figure 7) and

petrographic relationships which indicate an early-diagenetic origin (Figure 5 and Supplementary Material).

However, the Lower Velkerri Formation that is within the extent of this secondary overprint yielded younger Rb–Sr

ages (Figure 6) consistent with the age of the Derim Derim Dolerite (Ahmad and Munson, 2013; Bodorkos et al.,

2020; Nixon et al., 2021; Yang et al., 2020). Samples from this subset also recorded perturbed trace element

signatures (Figure 7) as well as fissile, foliated, and crystalline morphologies (Figure 5 and Supplementary

Material). Importantly, this corresponds with disturbed thermal maturity indicators (Figure 2, 3, and 4), suggesting

that the system is stable up to the maturation oil window and reset when the kerogen is overmature. Thermal

modelling of the Derim Derim Dolerite suggests that a 75 m thick intrusion at the base of the Altree 2 well would

have significantly elevated temperatures within 800 m of the sill, driving kerogen into the gas window, mobilising

trace elements and resetting the Rb–Sr isotopic system in the Lower Velkerri Formation.

Furthermore, thermal maturation indicators show that the Lower Verlkerri Formation experienced

palaeotemperatures of at least 110°C (Dellisanti et al., 2010; Espitalié, 1986; Frey and Merriman, 1999; Hunt, 1995;

Welte and Tissot, 1984). Along with elevated temperature, evidence of fluid-driven reactions is invoked as an

important process in the perturbation of these shales. Consequently, we show that the *in situ* Rb–Sr dating of the

Velkerri Formation combined with common hydrocarbon maturity proxies can help reveal the thermochronological

history of Proterozoic argillaceous rocks (Figure 9). When used in tandem, these methods can constrain the age of

deposition as well as subsequent secondary, late-stage geological events. Importantly, we demonstrate that this

technique can aid sedimentary-hosted resource exploration, as hydrothermal overprints can be identified and dated

as previously demonstrated in Subarkah et al. (2021). Specifically, for hydrocarbon exploration, we show that the





thermo-kinetic parameters of shale-hosted Rb–Sr isotopic system in hydrothermal settings can coincide with the maturation of kerogen into the gas window (Dodson, 1973; Espitalié, 1986; Kubler, 1967).

**9 Figure Captions**

**Figure 1A, left. Schematic stratigraphy and geochronological summary of the Roper Group (Abbott et al., 2001; Jackson et al., 1999; Kendall et al., 2009; Southgate et al., 2000; Subarkah et al., 2021; Yang et al., 2020). B, right. Sample location and depth to basement map for the McArthur Basin adapted from Geoscience (2018).**

**Figure 2. Summary of reprocessed down-hole well log data for Altree 2.**

**Figure 3. Covariation between $T_{max}$ values from pyrolysis analysis and illite crystallinity KI in the Velkerri Formation. An increase in $T_{max}$ coincide with a decrease in KI, suggesting that these proxies are both mainly sensitive to changes in palaeotemperature.**

**Figure 4. Kerogen characterisation of the Velkerri Formation based on organic geochemical parameters (Espitalié et al., 1985; Espitalié et al., 1977). Note that the Lower Velkerri Formation primarily hosts gas-prone kerogen. Upper Velkerri Formation = blue, Middle Velkerri Formation = teal, Lower Velkerri Formation = pink.**

**Figure 5. Spectral reflectance MLA maps of samples selected for *in situ* laser ablation analysis in this study overlain on top of their respective BSE images.**

**Figure 6. Summary of *in situ* Rb–Sr geochronological results from this study. Note that the best dated samples have the better spread in $^{87}Rb/^{86}Sr$ ratios, as suggested by Nebel (2014).**

**Figure 7. Statistical relationships between alteration proxies obtained from this study through laser ablation analysis (A) and whole-rock geochemical data (B) compiled from Cox et al. (2016).**

**Figure 8A. One-dimensional thermal model for sill intrusion of 75 m thickness within the Altree 2 well depicting time steps following emplacement at 0 ka. Sill intrusion and Rb–Sr sample depths have been normalised to palaeodepths with 1 km of additional Mesoproterozoic sediments. Median palaeotemperature estimates from $T_{max}$ data from the Altree 2 well have been included for comparison to modelled temperatures. B. Time-temperature profile for sample intervals within the Altree 2 well following intrusions of a sill of 75 m thick.**

**Figure 9. Summary of the thermochronological history of the Velkerri Formation in Altree 2. Black bars = *in situ* Rb–Sr shale ages. Re–Os shale geochronology from Kendall et al. (2009) and U–Pb dating of the Derim Derim Dolerite intrusion from Yang et al. (2020). Dashed lines = compilation of $T_{max}$ data in this study (Cox et al., 2016; NTGS, 2009, 2010).**

**10 Acknowledgments**

This work was supported by the Australian Research Council Projects LP160101353 and LP200301457 with Santos Ltd, Empire Energy Group Ltd, Northern Territory Geological Survey and Origin as partners. The initial





development and validation of *in situ* Rb–Sr dating technique at the University of Adelaide was also supported by

Agilent Technologies Australia Ltd. This forms MinEx CRC contribution #2022/XX. Aoife McFadden is thanked

for their assistance in the MLA mapping of the samples in this study.

## 11 Author Contributions

Darwinaji Subarkah (primary author): Conceptualisation, method development, experimentation, manuscript

drafting.

Angus Leslie Nixon: Conceptualisation, computational modelling, manuscript drafting.

Monica Jimenez: Conceptualisation, manuscript drafting.

Alan Stephen Collins: Conceptualisation, primary supervision, manuscript drafting.

Morgan Lee Blades: Sampling, method development, experimentation, manuscript drafting, secondary supervision.

Juraj Farkaš: Conceptualisation, method development, secondary supervision.

Sarah Gilbert: Method development, experimentation, manuscript drafting.

Simon Holford: Manuscript drafting.

## 12 Competing Interests

The authors declare that they have no conflict of interest.

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

        Contributions to Mineralogy and Petrology, v. 40, no. 3, p. 259-274.
Duddy, I., Green, P., Gibson, H., and Hegarty, K., 2004, Regional Palaeothermal episodes in
Northern Australia: Timor Sea Petrol. Geosci. (Proc. Timor Sea Symp. 2003).
Dutkiewicz, A., Volk, H., Ridley, J., and George, S. C., 2004, Geochemistry of oil in fluid
        inclusions in a middle Proterozoic igneous intrusion: implications for the source of
        hydrocarbons in crystalline rocks: Organic Geochemistry, v. 35, no. 8, p. 937-957.
Eberl, D., and Velde, B., 1989, Beyond the Kubler index: Clay minerals, v. 24, no. 4, p. 571-577.
Espitalié, J., 1986, Use of Tmax as a maturation index for different types of organic matter:
        comparison with vitrinite reflectance: Collection colloques et séminaires-Institut français
        du pétrole, no. 44, p. 475-496.
Espitalié, J., Deroo, G., and Marquis, F., 1985, La pyrolyse Rock-Eval et ses applications.
        Deuxième partie: Revue de l'Institut français du Pétrole, v. 40, no. 6, p. 755-784.





Espitalié, J., Madec, M., Tissot, B., Mennig, J., and Leplat, P., Source rock characterization method for petroleum exploration, *in* Proceedings Offshore Technology Conference1977, OnePetro.

Evins, L. Z., Jourdan, F., and Phillips, D. J. L., 2009, The Cambrian Kalkarindji Large Igneous Province: Extent and characteristics based on new 40Ar/39Ar and geochemical data, v.

110, no. 1-4, p. 294-304.

Faure, G., 1977, Principles of isotope geology.

Field, D., and Råheim, A., 1979, A geologically meaningless Rb–Sr total rock isochron: Nature, v. 282, no. 5738, p. 497-499.

Frey, R. M. M., and Merriman, R., 1999, Patterns of very low-grade metamorphism in

metapelitic rocks M: Frey D. Robinson Low-Grade Metamorphism Blackwell Science Oxford, v. 61, p. 107.

Galán, E., 2006, Genesis of clay minerals: Developments in clay science, v. 1, p. 1129-1162.

Gard, M., Hasterok, D., and Halpin, J. A. J. E. S. S. D., 2019, Global whole-rock geochemical database compilation, v. 11, no. 4, p. 1553-1566.

George, S., and Ahmed, M., 2002, Use of aromatic compound distributions to evaluate organic maturity of the Proterozoic middle Velkerri Formation, McArthur Basin, Australia.

Geoscience, F. J. N. T. G. S., Digital Information Package, DIP, 2018, SEEBASE® study and GIS for greater McArthur Basin, v. 17.

Glass, L. M., and Phillips, D. J. G., 2006, The Kalkarindji continental flood basalt province: A

new Cambrian large igneous province in Australia with possible links to faunal extinctions, v. 34, no. 6, p. 461-464.

Gorojovsky, L., and Alard, O., 2020, Optimisation of laser and mass spectrometer parameters for the in situ analysis of Rb/Sr ratios by LA-ICP-MS/MS: Journal of Analytical Atomic Spectrometry, v. 35, no. 10, p. 2322-2336.

Govindaraju, K., Rubeska, I., and Paukert, T., 1994, 1994 Report On Zinnwaldite Zw-C Analysed By Ninety-Two Git-Iwg Member-Laboratories: Geostandards Newsletter, v. 18, no. 1, p. 1-42.

Guggenheim, S., Bain, D. C., Bergaya, F., Brigatti, M. F., Drits, V. A., Eberl, D. D., Formoso, M. L., Galán, E., Merriman, R. J., and Peacor, D. R., 2002, Report of the Association

Internationale pour l'Etude des Argiles (AIPEA) Nomenclature Committee for 2001: order, disorder and crystallinity in phyllosilicates and the use of the 'crystallinity index': Clay Minerals, v. 37, no. 2, p. 389-393.

Hahn, O., Strassman, F., Mattauch, J., and Ewald, H., 1943, Geologische Altersbestimmungen mit der strontiummethode: Chem. Zeitung, v. 67, p. 55-56.

Hahn, O., and Walling, E., 1938, Über die Möglichkeit geologischer Altersbestimmungen rubidiumhaltiger Mineralien und Gesteine: Zeitschrift für anorganische und allgemeine Chemie, v. 236, no. 1, p. 78-82.

Hall, L., Boreham, C. J., Edwards, D. S., Palu, T., Buckler, T., Troup, A., and Hill, A., 2016, Cooper Basin Source Rock Geochemistry, Geoscience Australia.

Harrison, T. M., Heizler, M. T., McKeegan, K. D., and Schmitt, A. K., 2010, In situ 40K–40Ca 'double-plus' SIMS dating resolves Klokken feldspar 40K–40Ar paradox: Earth and Planetary Science Letters, v. 299, no. 3-4, p. 426-433.

Hillier, S., 1995, Erosion, sedimentation and sedimentary origin of clays, Origin and mineralogy of clays, Springer, p. 162-219.





Hogmalm, K. J., Dahlgren, I., Fridolfsson, I., and Zack, T., 2019, First in situ Re-Os dating of
          molybdenite by LA-ICP-MS/MS: Mineralium Deposita, v. 54, no. 6, p. 821-828.

Hogmalm, K. J., Zack, T., Karlsson, A. K. O., Sjöqvist, A. S. L., and Garbe-Schönberg, D.,
          2017, In situ Rb–Sr and K–Ca dating by LA-ICP-MS/MS: an evaluation of N2O and SF6
          as reaction gases: Journal of Analytical Atomic Spectrometry, v. 32, no. 2, p. 305-313.

Hunt, J. M., 1995, Petroleum geochemistry and geology.

Isson, T. T., and Planavsky, N. J., 2018, Reverse weathering as a long-term stabilizer of marine
          pH and planetary climate: Nature, v. 560, no. 7719, p. 471-475.

Iyer, K., Svensen, H., and Schmid, D. W., 2018, SILLi 1.0: a 1-D numerical tool quantifying the
          thermal effects of sill intrusions: Geosci. Model Dev., v. 11, no. 1, p. 43-60.

Jackson, M., Sweet, I., Page, R., and Bradshaw, B., 1999, The South Nicholson and Roper
          Groups: evidence for the early Mesoproterozoic Roper Superbasin: Integrated Basin
          Analysis of the Isa Superbasin using Seismic, Well-log, and Geopotential Data: An
          Evaluation of the Economic Potential of the Northern Lawn Hill Platform: Canberra,
          Australia, Australian Geological Survey Organisation Record, v. 19.

Jackson, M. J., Muir, M. D., Plumb, K. A., Australia. Bureau of Mineral Resources, G., and
          Geophysics, 1987, Geology of the Southern McArthur Basin, Northern Territory,
          Australian Government Pub. Service.

Jarrett, A. J., Cox, G. M., Brocks, J. J., Grosjean, E., Boreham, C. J., and Edwards, D. S., 2019,
          Microbial assemblage and palaeoenvironmental reconstruction of the 1.38 Ga Velkerri
Formation, McArthur Basin, northern Australia: Geobiology, v. 17, no. 4, p. 360-380.

Jarvie, D. M., 1991, Factors affecting Rock-Eval derived kinetic parameters: Chemical Geology,
          v. 93, no. 1-2, p. 79-99.

Jenkin, G. R., Rogers, G., Fallick, A. E., and Farrow, C. M., 1995, Rb‧Sr closure temperatures
          in bi-mineralic rocks: a mode effect and test for different diffusion models: Chemical
Geology, v. 122, no. 1-4, p. 227-240.

Jochum, K., and Stoll, B., 2008, Reference materials for elemental and isotopic analyses by LA-
          (MC)-ICP-MS: Successes and outstanding needs: Laser ablation ICP-MS in the Earth
          sciences: Current practices and outstanding issues, 147-168 (2008), v. 40.

Jochum, K. P., Weis, U., Stoll, B., Kuzmin, D., Yang, Q., Raczek, I., Jacob, D. E., Stracke, A.,
Birbaum, K., Frick, D. A., Günther, D., and Enzweiler, J., 2011, Determination of
          Reference Values for NIST SRM 610–617 Glasses Following ISO Guidelines:
          Geostandards and Geoanalytical Research, v. 35, no. 4, p. 397-429.

Jochum, K. P., Willbold, M., Raczek, I., Stoll, B., and Herwig, K., 2005, Chemical
          Characterisation of the USGS Reference Glasses GSA-1G, GSC-1G, GSD-1G, GSE-1G,
BCR-2G, BHVO-2G and BIR-1G Using EPMA, ID-TIMS, ID-ICP-MS and
          LA-ICP-MS: Geostandards and Geoanalytical Research, v. 29, no. 3, p. 285-302.

Jourdan, F., Hodges, K., Sell, B., Schaltegger, U., Wingate, M., Evins, L., Söderlund, U., Haines,
          P., Phillips, D., and Blenkinsop, T. J. G., 2014, High-precision dating of the Kalkarindji
          large igneous province, Australia, and synchrony with the Early–Middle Cambrian (Stage
4–5) extinction, v. 42, no. 6, p. 543-546.

Kendall, B., Creaser, R., Gordon, G., and Anbar, A., 2009, Re-Os and Mo isotope systematics of
          black shales from the Middle Proterozoic Velkerri and Wollogorang Formations,
          McArthur Basin, northern Australia: Geochimica et Cosmochimica Acta, v. 73, p. 2534-
          2558.





Kennedy, M., Droser, M., Mayer, L. M., Pevear, D., and Mrofka, D., 2006, Late Precambrian oxygenation; inception of the clay mineral factory: Science, v. 311, no. 5766, p. 1446-1449.

Kosakowski, G., Kunert, V., Clauser, C., Franke, W., and Neugebauer, H. J., 1999, Hydrothermal transients in Variscan crust: paleo-temperature mapping and hydrothermal
models: Tectonophysics, v. 306, no. 3, p. 325-344.

Kubler, B., 1967, La cristallinité de l'illite et les zones tout à fait supérieures du métamorphisme: Etages tectoniques, p. 105-121.

Lanigan, K., and Ledlie, I. M., 1990, Walton-1,2 EP 24 McArthur Basin, Northern Territory Well Completion Report: Pacific Oil and Gas, PR1989-0088.
Lanigan, K., and Torkington, J., 1991, Well Completion Report EP19 - Sever 1, Daly Sub-basin of the McArthur Basin: Pacific Oil and Gas, PR1990-0069.

Laureijs, C. T., Coogan, L. A., and Spence, J., 2021, In-situ RbSr dating of celadonite from altered upper oceanic crust using laser ablation ICP-MS/MS: Chemical Geology, p. 120339.
Ledlie, I. M., and Maim, K., 1989, Lawrence 1 EP 5 McArthur Basin, Northern Territory Well Completion Report: Pacific Oil and Gas, PR1989-0005.

Lee, M., and Parsons, I., 1999, Biomechanical and biochemical weathering of lichen-encrusted granite: textural controls on organic–mineral interactions and deposition of silica-rich layers: Chemical Geology, v. 161, no. 4, p. 385-397.
Lemiux, Y., 2011, Altree 2, Burdo 1, Chanin 1, Jamison 1, McManus 1, Shenandoah 1A, Walton 2, Balmain-1, Elliott-1 pyrolysis and tight rock analysis: Talisman Energy,

Advanced Well Technologies, Northern Territory Geological Survey, CSR0192

Lev, S. M., McLennan, S. M., and Hanson, G. N., 1999, Mineralogic controls on REE mobility during black-shale diagenesis: Journal of Sedimentary Research, v. 69, no. 5, p. 1071-
1082.

Li, S.-S., Santosh, M., Farkaš, J., Redaa, A., Ganguly, S., Kim, S. W., Zhang, C., Gilbert, S., and Zack, T., 2020, Coupled U-Pb and Rb-Sr laser ablation geochronology trace Archean to Proterozoic crustal evolution in the Dharwar Craton, India: Precambrian Research, v. 343, p. 105709.
Li, S., Wang, X.-C., Li, C.-F., Wilde, S. A., Zhang, Y., Golding, S. D., Liu, K., and Zhang, Y., 2019, Direct Rubidium-Strontium Dating of Hydrocarbon Charge Using Small Authigenic Illitic Clay Aliquots from the Silurian Bituminous Sandstone in the Tarim Basin, NW China: Scientific Reports, v. 9, no. 1, p. 1-13.

Mackenzie, F. T., and Kump, L. R., 1995, Reverse weathering, clay mineral formation, and
oceanic element cycles: Science, v. 270, no. 5236, p. 586-586.

Mählmann, R. F., Bozkaya, Ö., Potel, S., Le Bayon, R., Šegvić, B., and Nieto, F., 2012, The pioneer work of Bernard Kübler and Martin Frey in very low-grade metamorphic terranes: paleo-geothermal potential of variation in Kübler-Index/organic matter reflectance correlations. A review: Swiss Journal of Geosciences, v. 105, no. 2, p. 121-
152.

McMahon, W. J., and Davies, N. S., 2018, Evolution of alluvial mudrock forced by early land plants: Science, v. 359, no. 6379, p. 1022-1024.

Mergelov, N., Mueller, C. W., Prater, I., Shorkunov, I., Dolgikh, A., Zazovskaya, E., Shishkov, V., Krupskaya, V., Abrosimov, K., and Cherkinsky, A., 2018, Alteration of rocks by



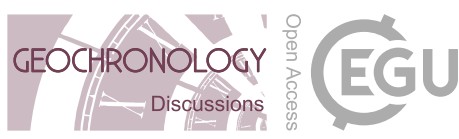

endolithic organisms is one of the pathways for the beginning of soils on Earth: Scientific reports, v. 8, no. 1, p. 1-15.

Meunier, A., Velde, B., and Velde, B., 2004, Illite: Origins, evolution and metamorphism, Springer Science & Business Media.

Minster, J. F., Ricard, L. P., and Alle`gre, C. J., 1979, 87Rb-87Sr chronology of enstatite
meteorites: Earth and Planetary Science Letters, v. 44, no. 3, p. 420-440.

Mukherjee, I., and Large, R. R., 2016, Pyrite trace element chemistry of the Velkerri Formation, Roper Group, McArthur Basin: Evidence for atmospheric oxygenation during the Boring Billion: Precambrian Research, v. 281, p. 13-26.

Munson, T., and Revie, D., 2018, Munson TJ and Revie D, 2018. Stratigraphic subdivision of the
Velkerri Formation, Roper Group, McArthur Basin, Northern Territory. Northern Territory Geological Survey, Record 2018-006.

Nebel, O., 2014, Rb–Sr Dating, Encyclopedia of Scientific Dating Methods, p. 1-19.

Nebel, O., Scherer, E. E., and Mezger, K., 2011, Evaluation of the 87Rb decay constant by age comparison against the U–Pb system: Earth and Planetary Science Letters, v. 301, no. 1,
p. 1-8.

Nguyen, K., Love, G. D., Zumberge, J. A., Kelly, A. E., Owens, J. D., Rohrssen, M. K., Bates, S. M., Cai, C., and Lyons, T. W., 2019, Absence of biomarker evidence for early eukaryotic life from the Mesoproterozoic Roper Group: Searching across a marine redox gradient in mid-Proterozoic habitability: Geobiology, v. 17, no. 3, p. 247-260.

Nixon, A. L., Glorie, S., Collins, A. S., Blades, M. L., Simpson, A., and Whelan, J. A., 2021, Inter-cratonic geochronological and geochemical correlations of the Derim Derim–Galiwinku/Yanliao reconstructed Large Igneous Province across the North Australian and North China cratons: Gondwana Research.

Norris, A., and Danyushevsky, L., 2018, Towards Estimating the Complete Uncertainty Budget
of Quantified Results Measured by LA-ICP-MS: Goldschmidt: Boston, MA, USA.

NTGS, 1989, Altree 1 and 2 EP 24 McArthur Basin, Northern Territory Well Completion Report, Pacific Oil and Gas.

-, 2009, Core Sample Analysis. Total Organic Carbon, Programmed Pyrolysis Data. Altree 2, Balmain 1, Elliott 1, Jamison 1, Core Sampling Reports: Northern Territory, Australia,
Falcon Oil & Gas, Weatherford Laboratories

-, 2010, EP24 Altree 2 Petrology and organic geochemistry: Eni Australia,
Geotechnical Services,
Falcon Oil & Gas,
Northern Territory Geological Survey, CSR0185.

-, 2012, Quantitative X-Ray Diffraction Analysis of 30 samples, *in* Survey, N. T. G., ed.: Northern Territory, Australia, Northern Territory Geological Survey.

-, 2014, Basic Well Completion Report, NT EP167, Tarlee S3: Pangaea Resources, PR2015-0016.

-, 2015, Basic Well Completion Report NT EP167 Birdum Creek 1: Pangaea Resources,
PR2016-W006.

-, 2016, Basic Well Completion Report NT - EP167 Wyworrie 1: Pangaea Resources, PR2016-W007.

Ola, P. S., Aidi, A. K., and Bankole, O. M., 2018, Clay mineral diagenesis and source rock assessment in the Bornu Basin, Nigeria: Implications for thermal maturity and source
rock potential: Marine and Petroleum Geology, v. 89, p. 653-664.





Olierook, H. K., Rankenburg, K., Ulrich, S., Kirkland, C. L., Evans, N. J., Brown, S., McInnes, B. I., Prent, A., Gillespie, J., and McDonald, B., 2020, Resolving multiple geological events using in situ Rb–Sr geochronology: implications for metallogenesis at Tropicana, Western Australia: Geochronology, v. 2, no. 2, p. 283-303.

Page, R. W., Jackson, M. J., and Krassay, A. A., 2000, Constraining sequence stratigraphy in north Australian basins: SHRIMP U–Pb zircon geochronology between Mt Isa and McArthur River*: Australian Journal of Earth Sciences, v. 47, no. 3, p. 431-459.

Papanastassiou, D. A., and Wasserburg, G. J., 1970, RbSr ages from the ocean of storms: Earth and Planetary Science Letters, v. 8, no. 4, p. 269-278.

Pearce, N. J., Perkins, W. T., Westgate, J. A., Gorton, M. P., Jackson, S. E., Neal, C. R., and Chenery, S. P., 1997, A compilation of new and published major and trace element data for NIST SRM 610 and NIST SRM 612 glass reference materials: Geostandards newsletter, v. 21, no. 1, p. 115-144.

Peters, K. E., 1986, Guidelines for evaluating petroleum source rock using programmed
pyrolysis: AAPG bulletin, v. 70, no. 3, p. 318-329.

Peters, K. E., and Cassa, M. R., 1994, Applied source rock geochemistry: Chapter 5: Part II. Essential elements.

Piedad-Sánchez, N., Izart, A., Martínez, L., Suárez-Ruiz, I., Elie, M., and Menetrier, C., 2004, Paleothermicity in the Central Asturian Coal Basin, North Spain: International Journal of
Coal Geology, v. 58, no. 4, p. 205-229.

Plumb, K., and Wellman, P., 1987, McArthur Basin, Northern Territory: mapping of deep troughs using gravity and magnetic anomalies: BMR Journal of Australian Geology & Geophysics, v. 10, no. 3, p. 243-251.

Poitrasson, F., Pin, C., and Duthou, J.-L., 1995, Hydrothermal remobilization of rare earth
elements and its effect on Nd isotopes in rhyolite and granite: Earth and Planetary Science Letters, v. 130, no. 1, p. 1-11.

Pollastro, R. M., 1993, Considerations and applications of the illite/smectite geothermometer in hydrocarbon-bearing rocks of Miocene to Mississippian age: Clays and Clay minerals, v. 41, p. 119-119.

Rafiei, M., and Kennedy, M., 2019, Weathering in a world without terrestrial life recorded in the Mesoproterozoic Velkerri Formation: Nature Communications, v. 10, no. 1, p. 3448.

Rafiei, M., Löhr, S., Baldermann, A., Webster, R., and Kong, C., 2020, Quantitative petrographic differentiation of detrital vs diagenetic clay minerals in marine sedimentary sequences: Implications for the rise of biotic soils: Precambrian Research, v. 350, p.
885   105948.

Rawlings, D. J., 1999, Stratigraphic resolution of a multiphase intracratonic basin system: the McArthur Basin, northern Australia: Australian Journal of Earth Sciences, v. 46, no. 5, p. 703-723.

Redaa, A., Farkaš, J., Gilbert, S., Collins, A. S., Wade, B., Löhr, S., Zack, T., and Garbe-
Schönberg, D., 2021a, Assessment of elemental fractionation and matrix effects during in situ Rb–Sr dating of phlogopite by LA-ICP-MS/MS: implications for the accuracy and precision of mineral ages: Journal of Analytical Atomic Spectrometry.

Redaa, A., Farkaš, J., Hassan, A., Collins, A. S., Gilbert, S., and Löhr, S. C., 2021b, Constraints from in-situ Rb-Sr dating on the timing of tectono-thermal events in the Umm Farwah
shear zone and associated Cu-Au mineralisation in the Southern Arabian Shield, Saudi Arabia: Journal of Asian Earth Sciences, p. 105037.



Revie, D., 2014, XRD analysis greater McArthur Basin, *in* Survey, N. T. G., ed.: Northern Territory, Australia, Northern Territory Geological Survey.

Revie, D., 2016, Interpretive summary of integrated petroleum geochemistry of selected wells in the greater McArthur Basin, NT, Australia: Northern Territory Geological Survey, Weatherford Laboratories, CSR0413.

Revie, D., and MacDonald, G., Volumetric resource assessment of the lower Kyalla and middle Velkerri formations of the McArthur Basin, *in* Proceedings Annual Geoscience Exploration Seminar (AGES) Proceedings2017, Volume 28, p. 29.

Ribeiro, B. V., Finch, M. A., Cawood, P. A., Faleiros, F. M., Murphy, T. D., Simpson, A., Glorie, S., Tedeschi, M., Armit, R., and Barrote, V. R., 2021, From microanalysis to supercontinents: Insights from the Rio Apa Terrane into the Mesoproterozoic SW Amazonian Craton evolution during Rodinia assembly: Journal of Metamorphic Geology, v. n/a, no. n/a.

Sander, R., Pan, Z., Connell, L. D., Camilleri, M., Grigore, M., and Yang, Y., 2018, Controls on methane sorption capacity of Mesoproterozoic gas shales from the Beetaloo Sub-basin, Australia and global shales: International Journal of Coal Geology, v. 199, p. 65-90.

Shepherd, T. J., and Darbyshire, D. P. F., 1981, Fluid inclusion Rb–Sr isochrons for dating mineral deposits: Nature, v. 290, no. 5807, p. 578-579.

Simmons, E. C., 1998, rubidiumRubidium: Element and geochemistry, Geochemistry: Dordrecht, Springer Netherlands, p. 555-556.

Simpson, A., Gilbert, S., Tamblyn, R., Hand, M., Spandler, C., Gillespie, J., Nixon, A., and Glorie, S., 2021, In-situ LuHf geochronology of garnet, apatite and xenotime by LA ICP MS/MS: Chemical Geology, v. 577, p. 120299.

Singer, A., 1980, The paleoclimatic interpretation of clay minerals in soils and weathering profiles: Earth-Science Reviews, v. 15, no. 4, p. 303-326.

Southgate, P. N., Bradshaw, B. E., Domagala, J., Jackson, M. J., Idnurm, M., Krassay, A. A., Page, R. W., Sami, T. T., Scott, D. L., Lindsay, J. F., McConachie, B. A., and Tarlowski, C., 2000, Chronostratigraphic basin framework for Palaeoproterozoic rocks (1730–1575 Ma) in northern Australia and implications for base-metal mineralisation: Australian Journal of Earth Sciences, v. 47, no. 3, p. 461-483.

Subarkah, D., Blades, M. L., Collins, A. S., Farkaš, J., Gilbert, S., Löhr, S. C., Redaa, A., Cassidy, E., and Zack, T., 2021, Unraveling the histories of Proterozoic shales through in situ Rb-Sr dating and trace element laser ablation analysis: Geology.

Summons, R. E., Taylor, D., and Boreham, C. J., 1994, GEOCHEMICAL TOOLS FOR EVALUATING PETROLEUM GENERATION IN MIDDLE PROTEROZOIC SEDIMENTS OF THE McARTHUR BASIN, NORTHERN TERRITORY, AUSTRALIA: The APPEA Journal, v. 34, no. 1, p. 692-706.

Tamblyn, R., Hand, M., Morrissey, L., Zack, T., Phillips, G., and Och, D., 2020, Resubduction of lawsonite eclogite within a serpentinite-filled subduction channel: Contributions to Mineralogy and Petrology, v. 175, no. 8, p. 74.

Tamblyn, R., Hand, M., Simpson, A., Gilbert, S., Wade, B., and Glorie, S., 2021, In situ laser ablation Lu–Hf geochronology of garnet across the Western Gneiss Region: campaign-style dating of metamorphism: Journal of the Geological Society.

Taylor, D., Kontorovich, A. E., Larichev, A. I., and Glikson, M., 1994, Petroleum Source Rocks In The Roper Group Of The Mcarthur Basin: Source Characterisation And Maturity



Determinations Using Physical And Chemical Methods: The APPEA Journal, v. 34, no. 1, p. 279-296.

Tillberg, M., Drake, H., Zack, T., Kooijman, E., Whitehouse, M. J., and Åström, M. E., 2020, In situ Rb-Sr dating of slickenfibres in deep crystalline basement faults: Scientific Reports, v. 10, no. 1, p. 562.

Tissot, B., Durand, B., Espitalie, J., and Combaz, A., 1974, Influence of nature and diagenesis of organic matter in formation of petroleum: Aapg Bulletin, v. 58, no. 3, p. 499-506.

Tissot, B., Pelet, R., and Ungerer, P., 1987, Thermal history of sedimentary basins, maturation indices, and kinetics of oil and gas generation: AAPG bulletin, v. 71, no. 12, p. 1445-1466.

Torgersen, E., Viola, G., Zwingmann, H., and Harris, C., 2015, Structural and temporal evolution of a reactivated brittle–ductile fault – Part II: Timing of fault initiation and reactivation by K–Ar dating of synkinematic illite/muscovite: Earth and Planetary Science Letters, v. 410, p. 212-224.

Varajao, A., and Meunier, A., 1995, Particle morphological evolution during the conversion of I/S to illite in Lower Cretaceous shales from Sergipe-Alagoas Basin, Brazil: Clays and Clay minerals, v. 43, no. 1, p. 14-28.

Velde, B., and Espitalié, J., 1989, Comparison of kerogen maturation and illite/smectite somposition in diagnesis: Journal of Petroleum Geology, v. 12, no. 1, p. 103-110.

Velde, B., and Vasseur, G., 1992, Estimation of the diagenetic smectite to illite transformation in time-temperature space: American Mineralogist, v. 77, no. 9-10, p. 967-976.

Vermeesch, P., 2018, IsoplotR : A free and open toolbox for geochronology: Geoscience Frontiers, v. 9.

Villa, 1998, Isotopic closure: Terra Nova, v. 10, no. 1, p. 42-47.

Villa, I. M., De Bièvre, P., Holden, N., and Renne, P., 2015, IUPAC-IUGS recommendation on the half life of 87Rb: Geochimica et Cosmochimica Acta, v. 164, p. 382-385.

Volk, H., George, S. C., Dutkiewicz, A., and Ridley, J., 2005, Characterisation of fluid inclusion oil in a Mid-Proterozoic sandstone and dolerite (Roper Superbasin, Australia): Chemical Geology, v. 223, no. 1, p. 109-135.

Waliczek, M., Machowski, G., Poprawa, P., Świerczewska, A., and Więcław, D., 2021, A novel VRo, Tmax, and S indices conversion formulae on data from the fold-and-thrust belt of the Western Outer Carpathians (Poland): International Journal of Coal Geology, v. 234, p. 103672.

Wang, X.-C., Li, Z.-X., Li, X.-H., Li, J., Liu, Y., Long, W.-G., Zhou, J.-B., and Wang, F. J. J. o. P., 2012, Temperature, pressure, and composition of the mantle source region of Late Cenozoic basalts in Hainan Island, SE Asia: a consequence of a young thermal mantle plume close to subduction zones?, v. 53, no. 1, p. 177-233.

Waples, D. W., 1980, Time and temperature in petroleum formation: application of Lopatin's method to petroleum exploration: AAPG bulletin, v. 64, no. 6, p. 916-926.

Warr, L., and Mählmann, R. F., 2015, Recommendations for Kübler index standardization: Clay Minerals, v. 50, no. 3, p. 283-286.

Warr, L., and Rice, A. J. J. o. m. G., 1994, Interlaboratory standardization and calibration of day mineral crystallinity and crystallite size data, v. 12, no. 2, p. 141-152.

Warren, J. K., George, S. C., Hamilton, P. J., and Tingate, P., 1998, Proterozoic Source Rocks: Sedimentology and Organic Characteristics of the Velkerri Formation, Northern Territory, Australia1: AAPG Bulletin, v. 82, no. 3, p. 442-463.



Welte, D., and Tissot, P., 1984, Petroleum formation and occurrence, Springer.

Williams-Jones, A., Migdisov, A., and Samson, I., 2012, Hydrothermal Mobilisation of the Rare Earth Elements - a Tale of "Ceria" and "Yttria": Elements, v. 8, p. 355-360.

Wilson, M. J., 1999, The origin and formation of clay minerals in soils: past, present and future perspectives: Clay minerals, v. 34, no. 1, p. 7-25.

Yang, B., Collins, A., Blades, M., Capogreco, N., Payne, J., Munson, T., Cox, G., and Glorie, S., 2019, Middle-late Mesoproterozoic tectonic geography of the North Australia Craton: U–Pb and Hf isotopes of detrital zircon grains in the Beetaloo Sub-basin, Northern Territory, Australia: Journal of the Geological Society, v. 176, p. jgs2018-2159.

Yang, B., Collins, A. S., Cox, G. M., Jarrett, A. J. M., Denyszyn, S., Blades, M. L., Farkaš, J., and Glorie, S., 2020, Using Mesoproterozoic Sedimentary Geochemistry to Reconstruct Basin Tectonic Geography and Link Organic Carbon Productivity to Nutrient Flux from a Northern Australian Large Igneous Province: Basin Research, v. n/a, no. n/a.

Yang, B., Smith, T. M., Collins, A. S., Munson, T. J., Schoemaker, B., Nicholls, D., Cox, G., Farkas, J., and Glorie, S., 2018, Spatial and temporal variation in detrital zircon age provenance of the hydrocarbon-bearing upper Roper Group, Beetaloo Sub-basin, Northern Territory, Australia: Precambrian Research, v. 304, p. 140-155.

Yang, S., and Horsfield, B., 2020, Critical review of the uncertainty of Tmax in revealing the thermal maturity of organic matter in sedimentary rocks: International Journal of Coal Geology, v. 225, p. 103500.

Yang, Y.-h., Zhang, H.-f., Chu, Z.-y., Xie, L.-w., and Wu, F.-y., 2010, Combined chemical separation of Lu, Hf, Rb, Sr, Sm and Nd from a single rock digest and precise and accurate isotope determinations of Lu–Hf, Rb–Sr and Sm–Nd isotope systems using Multi-Collector ICP-MS and TIMS: International Journal of Mass Spectrometry, v. 290, no. 2-3, p. 120-126.

Yim, S.-G., Jung, M.-J., Jeong, Y.-J., Kim, Y., and Cheong, A. C.-s., 2021, Mass fractionation of Rb and Sr isotopes during laser ablation-multicollector-ICPMS: in situ observation and correction: Journal of Analytical Science and Technology, v. 12, no. 1, p. 10.

Yoder, H. S., and Eugster, H. P., 1955, Synthetic and natural muscovites: Geochimica et Cosmochimica Acta, v. 8, no. 5, p. 225-280.

Zack, T., and Hogmalm, K. J., 2016, Laser ablation Rb/Sr dating by online chemical separation of Rb and Sr in an oxygen-filled reaction cell: Chemical Geology, v. 437, p. 120-133.

Zambell, C., Adams, J., Gorring, M., and Schwartzman, D., 2012, Effect of lichen colonization on chemical weathering of hornblende granite as estimated by aqueous elemental flux: Chemical Geology, v. 291, p. 166-174.

1025



# Figure 1

**A.**

**B.**



# Figure 2







**Figure 3**





# Figure 4



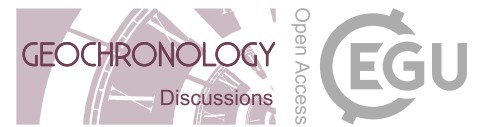

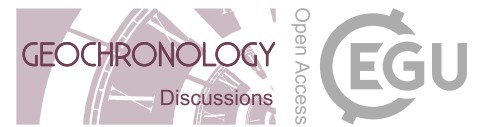

**Figure 5**



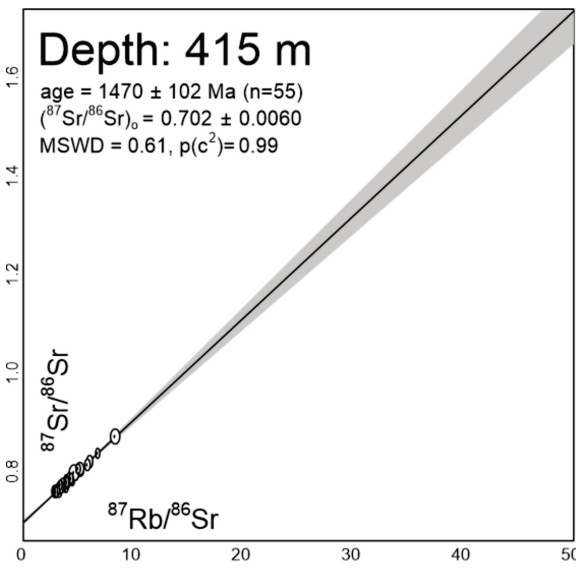

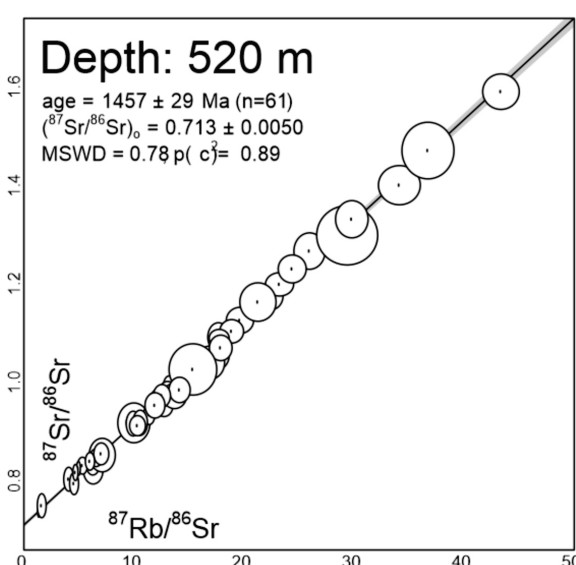

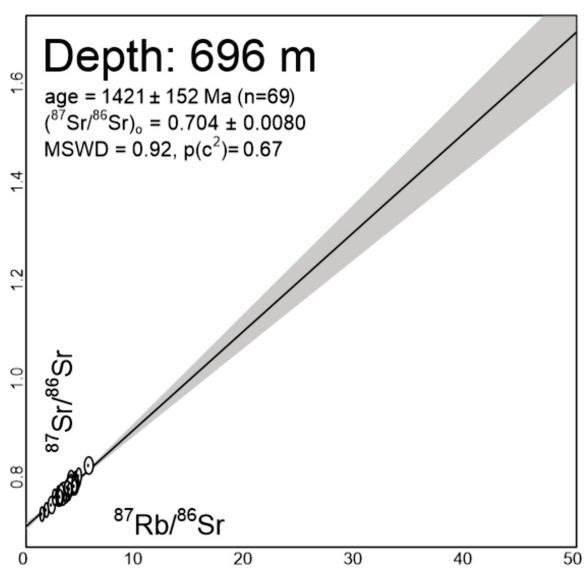

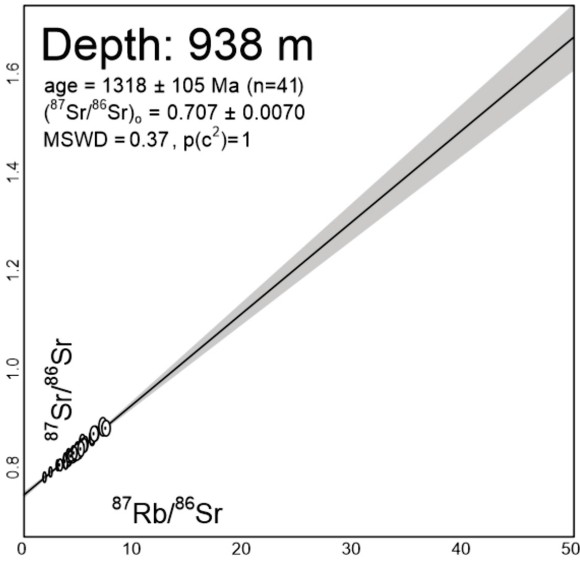

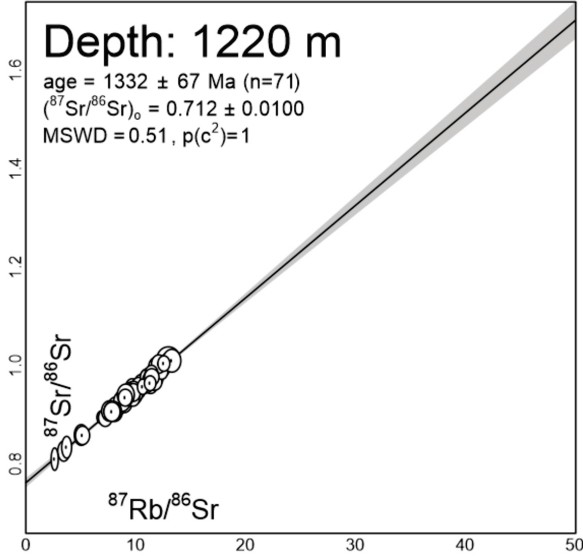



**Figure 7A.**






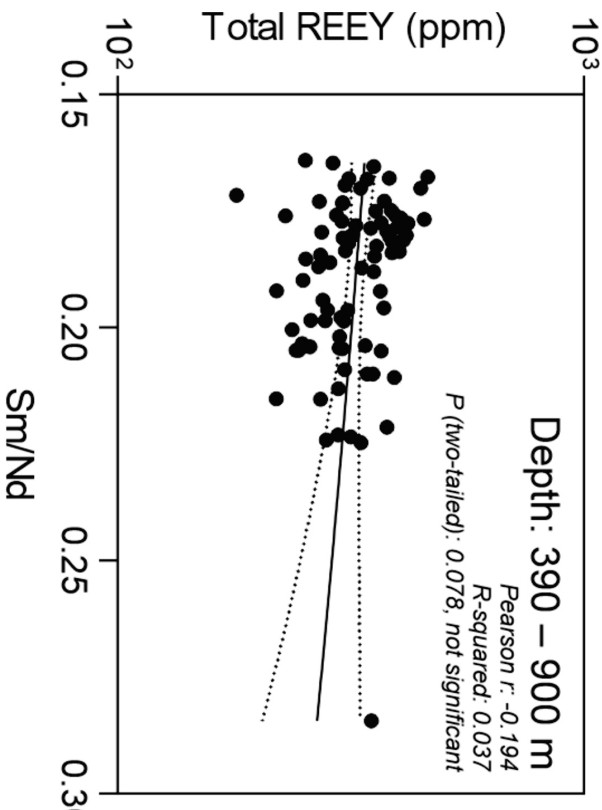

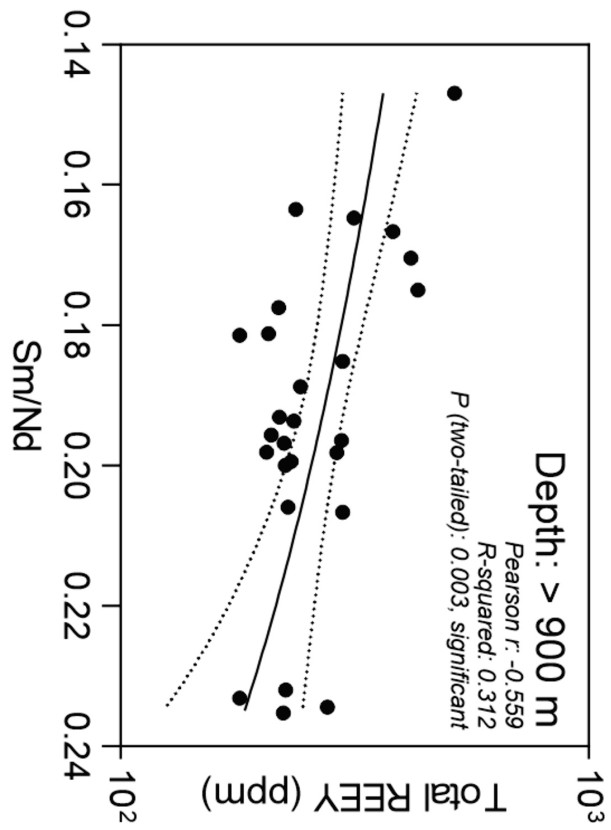

**Figure 7B.**



**Figure 8A.**

**Figure 8B.**





Figure 9