# Peer review of "Constraining the geothermal parameters of *in situ* Rb–Sr dating on Proterozoic shales and their subsequent applications"

_Geochronology, 2022_

## Author Comment (AC1)

**Geochronology Manuscript Comments from Referee 1:**

Subarkah et al. have investigated the links between burial history, palaeo-temperature evolution, illite formation age, illite crystallinity and organic matter maturation in Proterozoic shales from the McArthur Basin, Australia, by combining a literature study with novel petrographic analysis, thermal modelling of the effect of a dolerite intrusion and in-situ Rb–Sr dating of authigenic illitic clays via LA-ICP-MS/MS. The authors argue that the illite Rb-Sr system is stable at temperatures around the oil window but tends to be reset at higher temperatures around or above the gas window. The manuscript is well structured and the outcome is of potential interest for the geochemical society. Nonetheless, I have identified numerous (mostly minor) problems and inaccuracies that must be included before the manuscript can be accepted for publication. The present work would greatly benefit, if the authors:

1) Can provide vitrinite reflectance data for a direct temperature assessment?

Vitrinite is plant-derived and is only found in rocks from the Silurian and younger. There is no vitrinite in the Mesoproterozoic, and as such, this is not a tool at our disposal. However, we are able to calculate modelled Vitrinite Reflectance equivalents from our Tmax data following Jarvie et al., (2001). We will also be able to do this from new bitumen reflectance, methyl phenanthrene distribution factor and methyl phenanthrene ratio data collated from Jarret et al., (2019). We will include this as a new figure to show that the four thermal maturation indicators reflect the same elevated patterns down-hole and can also use this for a direct temperature assessment.

2) Could provide in-situ Rb-Sr glauconite ages (arguably the earliest diagenetic product) and compare with burial diagenetic illite formation ages?

Unfortunately, this is outside the scope of this study. More importantly, glauconite in samples here is also be a very minor component (1-2 wt. %), such that a laser spot target the phase will inherently incorporate the illite matrix within its vicinity. Furthermore, the illite-derived thermal constraints compiled in the study may also not reflect the formation of the glauconite phase in the samples.

3) Consider and discuss the possibility of a second pulse of illlite growth associated with the dolerite intrusion rather than a simple reset of the pristine signature of an inherited illite phase (pedogenic or early burial diagenetic), which would explain the decreasing Kübler indices with depth and the different illite morphologies.

Done. Some illite morphologies in the reset samples do look recrystallised and can be interpreted as a secondary growth. This description will be discussed further for further clarity.

4) Can identify (if possible) a relation between illite type (pore growth vs. lamellar), host rock lithology (shale vs. sandstone) and illite age (pristine vs. reset vs. newly formed). It looks like the sandstone-associated illite ages are reset (or represent illite neo-formation) while the shale-associated illite ages are true depositional or burial diagenetic ages. Is there a correlation between rock porosity, permeability, fluid transport and illite mineralization vs. resetting events?

We apologise for the mistake shown in Figure 2 as the samples are shown from the wrong depths here. The reset ages are samples from the Lower and Middle Velkerri Formation with a relatively rise in the gamma-ray well log which signify a relative increase in the grain size in

comparison to the Upper and Middle Velkerri Formation. However, the density and neutron well logs show similar values which represent similar porosity and permeability between the different formation intervals. Additionally, all of the samples analysed for Rb-Sr dating are shales, therefore we can corroborate that there are no distinguishing differences in lithology between the shale samples.

5) Can provide XRD patterns of illite and clay assemblages) to prove the absence of interstratified I-S in their samples?

 Done.

Taking all these aspects together, I recommend publication of the manuscript after some moderate revision with the aforementioned points in mind.

Other minor and major issues:

L20: "Hydrothermal" implies interaction of rocks with hot fluids generated from a cooling magma – better change to "thermal sensitivity".

 We disagree with the reviewer here, as hydrothermal only means "hot water" and does not suggest where that fluid comes from. We use the term "hydrothermal sensitivity" as the system is susceptible to fluid and heat. Changing this to "thermal sensitivity" would be incorrect and misleading on our behalf.

L22: Worthy to say that the study site is located in Australia…

 Done.

L27-29: There is something wrong with the sentence structure, please check.

 Done.

L39: "thermal systems absent of fluids" – this is extremely simplified; fluids are always present in shales given that hydrous phyllosilicates and organic matter are present.

 This will be rewritten for clarity.

L46-47: Worthy to say that distinctions between detrital and authigenic (plus resetting events) mineral phases must be made.

 Done.

L56: instruments // techniques

 'Instruments' is correct, as this is said in reference to the application of similar methods using different instruments such as a Multi-Collector.

L58: Define "reactive gas". // a LA-ICP-MS/MS system

 Done.

L62: I agree, but its worthy to mention that LA spot sizes (typically 50 or 75 µm) limit the area of investigation and that nm or µm sized mineral intergrowths of different origin are problematic, as they give mixed ages that need to be deconvoluted.

 Done.

L68: Clay minerals present in shales are barely visible with naked eyes, please re-phrase.

Done.

L71: …the samples analysed and of the fluids involved.

Done.

L72: has been

Done.

L78: Most clay mineral reactions take place under far-from-equilibrium conditions, especially in diagenetic settings, where, for instance, smectite matures to illite through mixed-layer I-S. This process can proceed over millions of years depending on T, K content, subsidence rate, fluid composition etc. (see Hower et al., 1976), and is never ever in equilibrium. Also, the term "water-column" is misleading here, as the seawater-derived fluids are always modified during diagenesis; better change to "burial fluids" or "formation waters" (or a similar phrase).

Done.

L81: …and erosion of soils and unstable parent rocks".

Done.

L90-98: I agree. However, it is worthy to say that illite maturation in shales via I-S formation is not an event-driven mineral reaction. Indeed, this process takes time (~Myr) and it is, until now, unclear how the Rb-Sr system in illite and I-S is affected during this continuous alteration process.

Done.

L99-101: Hard to follow, please simplify.

Done.

L102: Which type of clays is involved, illite, chlorite?

Illite.

L133: The saline Black Sea is characterized by a strong water-column redox stratification, while the Baltic Sea is brackish and mostly oxygenated (except for very deep basin parts, like Landsort deep). Perhaps the modern Black Sea is a better analogue for the Roper Group?

Both comparisons have been previously made in literature as seen in Ahmad and Munson (2013), Yang et al., (2020), and Cox et al., (2022).

L145: have been deposited? The U-Pb age of a detrital zircon cannot be used to establish a minimum age of a formation, because the zircons can record any age. However, I agree that the intrusion must be younger that the depositional age of the shale, so better say that the Kyalla Formation is somewhat older than 1313 Ma.

We disagree with the reviewer here. The Kyalla Formation cannot be older than the youngest detrital zircon age and is younger than the age of the intrusion. Therefore, the deposition window for the Kyalla Formation is bracketed between these two constraints.

L149: Depositional age or diagenetic age, given that we look at authigenic clays?

This was in relation to the Re-Os age referenced in literature, which has been interpreted as the depositional age. We will rewrite this for clarity.

L171: … which provide important complementary data to supplement this study, such as…

Done.

L182: Use of the word "sands". Isn`t it sandstone or weakly consolidated sandstone, given that metamorphic rocks (shale) are involved?

Done.

L188: collated // collected

Done.

L190: were // where

Done.

L191: at depths of

Done.

L195: Adelaide Microscopy – is this correct?

Yes.

L196: …acceleration voltage. MLA maps were collected

Done.

L199: Which chemical criteria were chosen to distinguish between smectite, illiite and I-S, or between different chlorite minerals? How good are the MLA quantifications compared to the bulk XRD data? Note here that Rafiei et al. (2020) report good data comparability for fine sized samples.

The minerals were identified through a comparison of the analysed spectra with a universal library provided by the Bruker AMICS software. The machine and program used by Rafiei et al. (2020) is different to the one used in this study and may result in a slight disparity.

L199-205: Were the Sr initials calculated from the isochrons or assumed to reflect Proterozoic seawater?

Calculated from the isochrons.

L214: The GLO age should be somewhat older (~100 Ma), because the host rock is of lower Cenomanian age. Please comment on this.

The age that we obtained (96 +/- 4 Ma) is accurate and within error to the published solution age of GL-O which is 94 +/- 1 Ma (Charbit et al., 1998 and Derkowski et al., 2009). This is noted as younger than a tuff-horizon U-Pb age for the unit dated at 113 +/- 0.3 Ma (Selby, 2009). As such, the ages obtained from GL-O have been proposed to instead either be indicative of the formation of glauconite in the host rock or the timing of isotopic closure of the mineral, occurring 4-5 m.y. after deposition (Selby, 2009 and Redaa et al., 2022). We will discuss this in the manuscript for clarity.

L229: Is there any mineralogical evidence for the assumed temperature of the sill, such as high-T mineral assemblages?

Sills of the Derim Derim Dolerite commonly comprise of plagioclase, clinopyroxene, hornblende, magnetite and minor quartz, although detailed analysis of the intrusion temperatures has not been conducted and is beyond the scope of this study. Sills are however interpreted as extracted from a mantle plume below the region (Yang et al., 2020; Nixon et al., 2021), and estimates of melt temperatures extracted from this source type have been used to constrain sill temperature (Wang et al., 2012).

L247: Tmax results compiled in this study range

Done.

L253: The Kübler Index is extremely high for true illite. Provide XRD patterns at EG-solvated state to determine the potential presence of I-S intermediates. If chlorite is present in the samples it is worthy to report the Archai Index as well, and cross-check with the Kübler Index data.

Done.

L257: If the temperature is > 120 °C kaolinite will change into dickite, please verify. Also, Mnt is not stable under these burial conditions. The only possibility of having Mnt in the samples is more recent sub-surface weathering (of for example feldspar).

Some clay phases have been identified as products of alteration of detrital feldspar. This is discussed in the Supplementary Material but we will discuss this further in the next iteration of the manuscript for clarity.

L260: Clinochlore is a Mg-rich chlorite – what is the difference between chlorite and clinochlore here?

The SEM instrument used here has differentiated between chlorite and clinochlore. Chlorite and clinochlore will be combined in the next iteration of the manuscript for clarity.

L263: XRD is not a destructive method by definition, but samples need to be crushed to obtain fine powders so that fabric information are lost, please revise.

Crushing the rock into powder for analysis destroys the petrographic relationship of the sample. After the sample is crushed for XRD, it cannot be reused for in situ analysis. We will rewrite this for clarity.

L264: Worthy to say that 1) MLA is often localized on small areas or layers that are not always representative for the bulk rock, 2) MLA assumes ideal mineral compositions and densities for mineral quantification and 3) MLA is based on 2D information. These aspects can make comparison with XRD datasets enigmatic.

Done.

L265: bulk XRD and MLA mapping results are summarized…

Done.

L271: Why is the age uncertainty so high in case of the shale samples investigated?

This is discussed in lines 272-278. Some samples simply do not have a wide range in Rb/Sr ratios, or are not abundant in Rb.

Table 1: The correlation between XRD and MLA datasets is OK (good) but far away from being consistent or excellent, as indicated by the authors. For example, kaolinite, montmorillonite and quartz are off by >10 wt.% in many cases, please clarify. Clinochlore is the Mg member of the chlorite family.

This is discussed in lines 260-265.

L313: Kübler index (KI) is determined by the 001-reflection of illite ….

Done.

L316: Vitrinite reflectance data can provide absolute formation temperatures.

As previously discussed, vitrinite is not present in the rock record during the Proterozoic. However, we will calculate vitrinite reflectance equivalents based on several maturity indicators such as Tmax, bitumen reflectance and aromatic hydrocarbon data in the next iteration of the manuscript.

L319: such as changes in heating rate

Done.

L335: …or possibly due to the presence of I-S in the samples? Provide XRD patterns for confirmation.

Done.

L376: formed within the host sediment during burial diagenesis

Done.

L377: A minor component of illite also replaces micas and feldspars…

Done.

L383: suggesting that the majority of illite formed relatively soon after sediment deposition.

Done.

L403: Unclear meaning of "more crystalline morphologies", re-phrase

Done.

L429: …recrystallized the former mineral assemblage or induced a second mineralization of clays.

Done.

L431-449: This is largely repetition, delete.

This paragraph is a summary of the discussions mentioned in the section and is used to conclude the argument. Hence, some points will be reiterated. However, we will rewrite this for clarity.

L456: Re-phrase: "which as intersected at present day depth 1696 m.".

Done.

L463: or induced a second mineralization event?

Done.

L496: crystalline illite morphologies

Done.

Figure 2: Why has TOC a negative value?

Done. Fixed figure.

Figure 5: Mineral coding is difficult to read, i.e. change mineral colors on the maps.

Done.

Figure 6: Can the authors explain the differences in the initial 87Sr/86Sr values among the sample set, i.e. ranging from radiogenic to seawater-type?

This is calculated from the isochrons. Each samples have a different spread in data, and as such, the initial $^{87}Sr/^{86}Sr$ value calculated from each isochron (i.e. the y-intercept) will differ.

Best regards

---

## Author Comment (AC2)

**Geochronology Manuscript Comments from Referee 2:**

The study uses in situ Rb-Sr dating of shale unit fragments and thermal modelling of literature data to constrain the thermochronological evolution with depth of an Australian Proterozoic basin site. It represents an important contribution towards increased understanding of the effects of heating and diffusion from secondary events on isotopic Rb-Sr systematics of basin rock units since few natural studies occur on the matter. The study also highlights the use for coupling knowledge of the thermal evolution in multiple dimensions when interpreting the significance and meaning of geochronological data. The utilized methodology offers a route to achieve that. However, the strength of each specific method and the combination of them is not demonstrated in much detail, as outlined by the following points.

1) The modelling of literature thermometry data with time is considerate and useful with vast amounts of input data, but since secondary mechanisms causing isotopic disturbance also can move laterally, the limit of a one-well model increases the interpretation uncertainty. In the absence of a horizontal modelling dimension, the following considerations should be discussed or clarified regarding the central issue of estimating the boundary conditions isotopic disturbance:

a) How may any lateral variations in the geological setting and the processes and events affecting the rock sequence impact the conclusions drawn on the timing, spatial occurrence and sheer cause of isotopic disturbance given that conclusions are based on a modelled vertical line?

      The vast majority of the thermal regime is controlled in the vertical dimension, which in this case is also the dimension in which the best geological constraints are present. Geologically, lateral transfer of heat is minimal, hence the prevalence of 1D models (e.g. Hall et al., 2020). Additionally, modelling in 2D or 3D space would require robust knowledge of the lateral geology which is not available such that it would improve reliability of our models. The only lateral thermal scenario which would significantly impact this model would be the emplacement of a (very) nearby intrusion at modelled stratigraphic levels which would increase temperatures broadly across the vertical sequence. Given there is no evidence for such a proximal intrusion, and thermometers from this well are adequately explained by the currently proposed overburden and bottom-hole intrusion (Figure 7), we suggest the current 1D model is appropriate. Furthermore, no major aquifers have been identified in the Velkerri Formation, suggesting that the potential for lateral fluid flow is unlikely.

b) Migrating fluids are inferred as cause for isotopic resetting beneath 900m depth. Can these fluids be traced by veins, mineralizations, crystal zonations or else? If so,

can direct thermometry or other geochemical characterization be applicable of such? Has this been observed and considered in any previous study of the site? Indifferent of negative or positive answers to these questions, the matter should be mentioned in the manuscript.

Migrating fluids have been observed by oil inclusions in veins cross-cutting the Derim Derim, Bessie Creek Sandstone, as reported in previous studies (Volk et al., 2005; Dutkiewicz et al., 2004). The petrographic textures of the reset shales are also more crystalline when compared to the unreset samples (now provided high-res SEM images in the revision).

2) The in situ Rb-Sr dating is the only new data collected in the study, and the technique is indeed promising and applicable for dating of diverse processes affecting shale units. In order for this analytical campaign to demonstrate the utility of the method, improvements in sample selection, presentation and discussion are due as outlined in the following points:

a) Given the spot size used and the fine-grained nature of illite, each isochron point represent a mixture of grains that may have stabilized isotopically at different times. The authors mention that each sample is predominantly composed of either unaffected or reset authigenic illite as observed by mineralogy alone. If XRD has been used to identify these in this or previous studies of the site, please provide and explain more explicitly the basis of the illite generation identifications. Is it verified that no mixtures of clay mineral phases or multiple clay growth generations (including detrital) are present in any of the samples? If so, how was this verified? If not, please comment on the implications for the age results and the interpretation of its meaning that multiple generations may exist. Please also clarify reasons for excluding grain size separation and Illite Age Analysis (Pevear, 1999) from the study.

We have now provided high resolution images to confirm lack of detrital clay minerals in samples, and elaborated on how such input can affect the resulting ages accordingly. SEM images were used to avoid coarser/more detrital-dominated bedding layers. For reset samples, all illite components (detrital and non-detrital) were assumed to be reset by the Derim Derim Dolerite intrusion. Trace element compositions (Zr, Si, Ti, REEY content) of each spot were also checked to filter detrital component.

Grain separation and illite age analysis destroys petrographic context, and can result in mixed ages. It is also time-consuming relative to the laser Rb-Sr method. This approach also cannot provide additional geochemical information (i.e. major and trace element data) which is an important deficit as it can be difficult to interpret the resulting age if mixed components do occur without these data.

b) Continuing on the illite mixing topic, estimates of the illite homogeneity can also

be provided by dissecting isotopic ratios in each LA spot signal in the absence of grain size separation. Please provide a detailed account on how the procedure of analysing spot homogeneity was carried out, on the outcomes and conclusions drawn from the observations, and mention any implications for the age results going from single downspot time frames to the combination of spots in the isochron diagram.

During processing, each spot was filtered by filtering bad signals (Si, Zr, not stable signals etc.). We like the idea of investigating single-spot isotopes variations for future investigation, but suggest that this is not required here. We calculated single-spot ages to further confirm that each spot in each sample consisted of clay phases that might not be homogeneous, but still form at the same time. This heterogeneity can actually be a positive, providing a good spread in the isochron and resulting in better errors.

c) Relating to the above points, inclusion of other minerals than illite in the LA signals is mentioned and disregarded as merely quartz in the Supplementary Material and therefore irrelevant for the Rb-Sr contents. Relating to the multimineral mineralogy maps, have the signals and spot locations been checked for mineral occurrence? If so, how was this performed? Were any spots rejected for the sake and if so, for what reasons? It was made sure that no K-bearing minerals interfered down-hole in each spot? Clarifying notes of these procedures and results should be contained at least in the supplementary information.

Similar point to previous comment. We have now elaborated further on how data processing is done (signal picking and cropping, checking major and trace elements for etc.) to clarify this.

d) the illite crystal textures and intermineralic textural relationships are qualitatively described without detailed petrographic images or accounts on variations within samples. Can such be added for the specific samples in this study and compared to previous studies describing these features at the site?

Similar point to previous comment again, we have provided more zoomed-in images and elaborate on petrographic content. And also provided additional comments and comparisons of this to Rafiei et al. (2019) and Subarkah et al., (2021) who worked on high-resolution petrography of Roper Group shales elsewhere.

e) The in situ Rb-Sr dating sample set consists of five samples over a ca. 800m depth interval. Given the discrepancy of several hundreds of meters (most shallow effect from the sill is interpreted at 600m or 800m) in the different thermal modelling predictions, please comment on how the sample interval larger than 200m below

696m depth affects the interpretation and uncertainty of the results regarding potential isotopic disturbance of fluid migration.

Four different thermal indicators (Tmax, two different aromatic hydrocarbons, bitumen reflectance, see Jarret et al., 2019) suggest that the elevated thermal gradient occurs past 900 m depths. Although the sub-sample set for the Rb-Sr analyses are sparse, the thermal data sample set are more continuous. Based on this and our thermal modelling, the isotopic disturbance should not occur prior to this depth.

f) The initial Sr values are not anchored to actual data but rather inferred from the isochrones and comes with large error ranges. Since the importance of initial Sr values for tracing crustal fluids and their sources is indeed stated in the manuscript, have any previous data source been considered for narrowing down on them in the stratigraphic sequence, or may new data collection on this be advisable? Given the spread in initial Sr values and their inference from large-error and low-Rb isochrons, it should be explicitly stated that the isochrons produce age errors ranging up to 300 Ma. Many of them overlap each other and several other dating results in the area, and yet their interpretation and meaning is provided without any note or disclaimer. The age errors and their implications for the conclusions based on the dating should be discussed. In addition, the concluding reasoning of the method as a useful discriminator of geological events in sedimentary units should regard the large age error ranges.

Good initial Sr values are limited to the availability of K(Rb)-deficient and Sr-bearing phases (e.g. carbonates) that form concurrently with the illite phases. Sometimes this is simply not available. Ideally, Sr data can be obtained from interbedded carbonates where possible. However, we haven't used the Sr initials from this study to make any interpretations. We also suggest that our method is less derivative as it calculates the initial ratio from the regression of the radiogenic Sr values and doesn't assume that other phases were cogenetic – it lets the illite data speak for itself!

The large errors for some samples is a fair point and hard to avoid in this technique. However data are separable and show meaningful and interpretable effects. This technique strives for accuracy over precision and we suggest that we have demonstrated the efficacy of it here. In addition, we will also attempt to provide the single spot ages for each sample. The population of single spot age results from the unaltered and altered samples should be statistically different from each other.

3) The combination of the methods have been shown to provide thermochronological constraints, but since the authors repeatedly emphasize its

utility, may it be described what actually makes this particular combination so powerful and how it distinguishes from other thermochronological methodological schemes?

Yes, we have now elaborated and compared this with other thermochronological methods. Primarily, other thermochron like K-Ar, Ar-Ar, fission tract (zircon, apatite) that date the surrounding sediments and make inferences on if this applies to the shales (or organic-rich units they want to constrain). They are also destructive to the petrographic context of each sample, time consuming, and expensive. On the other hand, traditional temperature constraints in petroleum systems (Tmax, vitrinite/bitumen reflectance, aromatic hydrocarbons) don't provide age data. This method allows for direct dating of these shales, and couple them with the thermal proxies previously mentioned. This is unique, fast and an affordable way to collect considerable useful data.

Specific comments

L161 Are there any tectono-thermal perturbation that would be expected to have affected the area, and if so, when and of what type? Any orogeny that may have disturbed radiogenic isotopic system?

Recent AFT thermochronology data presented by Nixon et al. (2022) across the McArthur Basin does indicate slow regional cooling during the Devonian-Ordovician Alice Springs Orogeny, attributed to minor regional uplift concurrent with this event. Crucially, there is no observed major structural reactivation within the basin associated with this event. This study does not find evidence for any other thermal perturbation within the basin following the Cambrian. Furthermore, no orogenic reworking is preserved in the McArthur Basin or regional basement in the form of metamorphism, igneous intrusion, large scale folding or angular unconformities, suggesting this region has not experienced major orogenesis following the Proterozoic

L167 Word missing after terminated? Otherwise the sentence does not make sense.

Will reword for clarity.

L280, L365 Avoid subjective adjectives such as good, here and on later places in the text.

Will reword for clarity.

L382-384 How can such a specific statement be motivated considering the large age errors?

Will reword for clarity, but also note that the data are distinguishable despite the errors

L456-458 Generally, chapter 5-8 contains multiple repetitions which can be slimmed. The sentence starting with "This event.." is one of those that includes statements already appearing repeatedly up to this point in the manuscript.

Will reword for clarity.

L488-501 Contains statements repeated from previous sections, but if this has the function of a concluding section it should work.

Will reword for clarity.

L493 Ages are in the text not seldom referred as comparative and relative, e.g. here in mentioning "younger ages". Precise age ranges would have made the text more concise and apt to follow in instances such as this.

Will reword for clarity.

Figure text Figure 6. Avoid the use of "better" and possibly the whole last sentence that can be deemed obvious and irrelevant.

Will reword for clarity.

Figure 5. The color scheme indeed needs adjustment, too many undistinguishable green colours.

Done.

Supplementary Material

L21 What is the last sentence supposed to mean? The signal interval selected in the data reduction procedure? Please clarify, and it is not helpful to put an explanatory word in apostrophes and then not explain it.

Done.

L45 Ideal is an interesting word of choice, would not more ideal for in situ spot-based LA dating at least be that individual grains can be targeted?

Will reword for clarity.

L56 Expressing that the textures do not look detrital should be replaced with a description and/or a detailed, high magnification petrographic photograph forming the basis of these genetic interpretations, which is also a general remark for the mineral texture descriptions (see comment 2d above).

> Will reword for clarity.

References

Pevear, D.R., 1999. Illite and hydrocarbon exploration. Proc. Natl. Acad. Sci. USA 96, 3440–3446. http://dx.doi.org/10.1073/pnas.96.7.3440.

---

## Author Response (AR1)

Dear Dr Subarkah

I recommend that you proceed with the revision based on the comments from the reviewers, along the lines of what detailed in your reply. Please make and effort to respond to as many comments as possible in the manuscript's main text, clarifying and adding information where needed.

In addition to the comments from the reviewers, I ask you to address the following points:

1. In methods, even if referring to a previous publications for the general set up, you should report the specific analytical conditions of those parameter that can change significantly from session to session, such as repetition rate, fluency, gas flow, spot size etc…

> This is done in the Supplementary Information.

2. Improve figure 6 by (i) zooming into the data part in each of them rather than plot all diagrams with the same scale; (ii) increase the font size of the number on axis and (iii) ensure that Sr initial value and its uncertainty are quoted with the same number of digits (0.702 ± 0.006, and not ±0.0060).

> Done.

3. None of the data plotted in figure 3, 4, 7 and 8A has uncertainties, for example for T, Sm/Nd, KI and other indexes. This suggests that either the uncertainty could not be calculated or it is smaller than the symbol. Please clarify in the caption which is the case and add an uncertainties wherever possible (an "typical uncertainty" cross at the corner of a diagram can also be used).

> When they are not plotted in the figures, uncertainties are smaller than the symbol. However, they can be found in the Supplementary Information.

4. The advantage of in situ analysis is to show textural context, so it would be better to show the location of the laser spots on the images of figure 5. This would also show clearly your spatial resolution compared to the fine grain of the samples.

> Done.

5. In the supplementary tables be aware of significant digits, particularly in Table S4 and LAICPMS element data. Reduce number to only significant digits.

> Done.

6. For clarity, repeat title and authors of the paper in the Supplementary Material file.

> Done.

Best regards
Daniela Rubatto
Associate Editor